# Brown and beige adipose tissue regulate systemic metabolism through a metabolite interorgan signaling axis

Anna Whitehead [1,7], Fynn N. Krause [2,7], Amy Moran[1,7], Amanda D. V. MacCannell[1], Jason L. Scragg[1], Ben D. McNally[2], Edward Boateng[1], Steven A. Murfitt[2], Samuel Virtue[3], John Wright[1], Jack Garnham[1], Graeme R. Davies[4], James Dodgson[5], Jurgen E. Schneider[1], Andrew J. Murray [6], Christopher Church[4], Antonio Vidal-Puig [3], Klaus K. Witte [1], Julian L. Griffin[2] & Lee D. Roberts [1✉]

Brown and beige adipose tissue are emerging as distinct endocrine organs. These tissues are functionally associated with skeletal muscle, adipose tissue metabolism and systemic energy expenditure, suggesting an interorgan signaling network. Using metabolomics, we identify 3-methyl-2-oxovaleric acid, 5-oxoproline, and β-hydroxyisobutyric acid as small molecule metabokines synthesized in browning adipocytes and secreted via monocarboxylate transporters. 3-methyl-2-oxovaleric acid, 5-oxoproline and β-hydroxyisobutyric acid induce a brown adipocyte-specific phenotype in white adipocytes and mitochondrial oxidative energy metabolism in skeletal myocytes both in vitro and in vivo. 3-methyl-2-oxovaleric acid and 5-oxoproline signal through cAMP-PKA-p38 MAPK and β-hydroxyisobutyric acid via mTOR. In humans, plasma and adipose tissue 3-methyl-2-oxovaleric acid, 5-oxoproline and β-hydroxyisobutyric acid concentrations correlate with markers of adipose browning and inversely associate with body mass index. These metabolites reduce adiposity, increase energy expenditure and improve glucose and insulin homeostasis in mouse models of obesity and diabetes. Our findings identify beige adipose-brown adipose-muscle physiological metabokine crosstalk.

[1] School of Medicine, University of Leeds, Leeds, UK. [2] Department of Biochemistry, University of Cambridge, Cambridge, UK. [3] Institute of Metabolic Science, University of Cambridge, Cambridge, UK. [4] Bioscience Metabolism, Research and Early Development, Cardiovascular, Renal and Metabolism, BioPharmaceuticals R&D, AstraZeneca, Cambridge, UK. [5] Phenotypic Screening and High Content Imaging, Antibody Discovery & Protein Engineering, R&D, AstraZeneca, Cambridge, UK. [6] Department of Physiology, Development and Neuroscience, University of Cambridge, Cambridge, UK. [7] These authors contributed equally: Anna Whitehead, Fynn N. Krause, Amy Moran. ✉email: L.D.Roberts@leeds.ac.uk

Brown adipose tissue (BAT) functions to regulate body temperature through non-shivering thermogenesis; the dissipation of chemical energy to produce heat[1,2]. Beige adipocytes are interspersed within the white adipose tissue (WAT) of rodents and humans, and can be induced to switch from a white-adipocyte-like phenotype to a brown-adipocyte-like phenotype; a process known as browning[3]. Brown and activated beige cells are characterized by high levels of fatty acid β-oxidation, mitochondrial content, and thermogenesis[4]. Thermogenesis occurs through the activity and increased expression of several specific gene products, including uncoupling protein 1 (UCP1), an inner mitochondrial membrane protein that uncouples substrate oxidation from ATP synthesis to generate heat[5]. Activated brown and beige adipocytes alter systemic energy metabolism, increasing substrate oxidation and energy expenditure, with potential for therapeutic exploitation for metabolic diseases including Type 2 diabetes mellitus (T2DM) and obesity[6,7].

The effects of BAT and beige adipose tissue on energy balance may not solely depend on the action of UCP1. Activation of thermogenesis in brown and beige adipose tissues may lead to propagation of thermogenesis in surrounding adipocytes and distal adipose depots. Transplantation studies of both beige and brown fat in mice suggest that these tissues can signal to activate endogenous beige and brown fat and improve glucose homeostasis in skeletal muscle[8,9]. In murine models of both adipose tissue browning and increased BAT thermogenesis, fatty acid oxidation in skeletal muscle is increased[8–10]. The anti-obesity and anti-diabetic effects of brown and beige adipose tissues are also not solely reliant on the thermogenic process. Mice lacking Ucp1 are resistant to diet-induced obesity at room temperature, yet mice lacking brown/beige fat are highly susceptible to an obese phenotype[11–15]. Therefore, beige and brown fat may influence systemic metabolism through non-UCP1 thermogenic mechanisms[16], potentially mediated through the release of endocrine signals in the adipocyte secretome.

In this study, we aimed to identify and characterize signals released in the browning adipocyte secretome that may influence systemic metabolism. A discrete panel of small molecule metabolite paracrine and endocrine signals, secreted from both beige and brown adipocytes, is identified. These metabokines increase adipose tissue, skeletal muscle, and systemic energy metabolism. We propose these brown and beige adipokine-like small molecules function in an adipose–adipose and adipose–skeletal muscle interorgan signaling axis.

## Results

### Metabolite signals from browning adipocytes increase brown-adipocyte-associated gene expression in primary adipocytes.
Adipocyte browning was induced in primary adipocytes differentiated from the stromal vascular fraction of subcutaneous (inguinal) WAT of mice using two distinct canonical signaling mechanisms, an adenylate cyclase activator (forskolin), and peroxisome proliferator-activated receptor δ (Pparδ) agonist (GW0742)[17]. Cells were washed and fresh serum-free media was conditioned on the cells for 24 h. Conditioned media was transferred to naïve primary adipocytes (Fig. 1a) and induced expression of brown-adipocyte-associated genes (Fig. 1b, c). Expression of brown-adipocyte-associated genes, including Ucp1, peroxisome proliferator-activated receptor γ co-activator1α (Pgc1α), cell death-inducing DFFA-like effector a (Cidea), carnitine palmitoyltransferase 1b (Cpt1b), acyl-CoA dehydrogenase very-long chain (Acadvl), and cytochrome C (Cycs) was further enhanced following media protein denaturation by boiling, implicating a nonprotein small molecule mediator(s) (Fig. 1b, c). These data may also indicate the presence of a secreted protein

inhibitor of browning. To define the physicochemical nature of the small molecule mediators, aqueous-soluble metabolites were extracted from media conditioned on activated beige adipocytes using solvent partition. The aqueous-soluble metabolites were reconstituted in fresh media and transferred to naïve primary adipocytes (Fig. 1d). Expression of brown-adipocyte-associated genes was induced by aqueous-soluble metabolites released from browning adipocytes.

To identify candidate metabolites that may induce browning, we applied both gas chromatography–mass spectrometry (GC-MS) and liquid chromatography–mass spectrometry (LC-MS) metabolic profiling to media conditioned on browning adipocytes. GW0742 and forskolin were not detected in conditioned media. Multivariate statistical models of the metabolic profiling data were used to identify common metabolite species enriched in the media by both cyclic AMP (cAMP) and PPARδ-induced browning (Fig. 1e). The concentration of sugar species and the branched-chain amino acids (BCAAs) valine and isoleucine was decreased in the media of browning adipocytes (Fig. 1f). Concomitantly the concentration of 5-oxoproline (5OP) and the BCAA catabolites α-hydroxyisocaproic acid (HIC), α-ketoisovaleric acid (AKV), α-hydroxyisovaleric acid (AHI), 3-methyl-2-oxovaleric acid (MOVA), β-hydroxyisobutyric acid (BHIBA), and β-hydroxyisovaleric acid (BHIVA) was increased in the media. Glycerol, a marker of lipolysis, was also increased.

Next, we examined whether physiological plasma concentrations of the BCAA metabolites and 5OP increased the expression of brown-adipocyte-associated genes in primary adipocytes (Fig. 1g)[18–21]. Physiological plasma concentrations of the metabolites are given in Supplementary Table 1. MOVA, 5OP, BHIBA, and BHIVA significantly and robustly induced expression of brown-adipocyte-associated genes including Ucp1, Cidea, and Cpt1b. MOVA, 5OP, BHIBA, and BHIVA were also enriched in the media of primary mouse canonical brown adipocytes following cAMP (forskolin 1 μM) or Pparδ (GW0742 100 nM)-mediated induction of brown-adipocyte-associated gene expression (Fig. 1h, i).

Therefore, the metabolites MOVA, 5OP, BHIBA, and BHIVA are released from primary white and brown adipocytes in response to thermogenic stimuli and induce the expression of brown-adipocyte-associated genes in naïve adipocytes.

### Metabolite signals are secreted from browning human adipocytes and induce a brown-adipocyte-like functional phenotype.
We determined whether the secretion of the candidate metabokines from browning adipocytes was conserved in human cells. A brown-adipocyte-like phenotype was induced in human primary adipocytes using either forskolin (1 μM) or a PPARδ agonist (100 nM GW0742) (Supplementary Fig. 1a–h). MOVA, 5OP, BHIBA, and BHIVA were enriched in the media of forskolin and PPARδ-agonist-treated human adipocytes (Fig. 2a). Treatment of primary human adipocytes with physiological plasma concentrations (Supplementary Table 1) of MOVA (20 μM), 5OP (20 μM), BHIBA (20 μM), and BHIVA (10 μM) induced expression of a panel of brown-adipocyte-associated genes including UCP1, CIDEA, and PGC1α (Fig. 2b). Induction of UCP1 expression in primary human adipocytes treated with metabolites in the physiological micromolar range occurred in a dose-dependent manner (Supplementary Fig. 2a–d). The concentrations of UCP1 protein in metabokine-treated human primary adipocytes were also increased (Fig. 2c). We further investigated whether the metabolites induced functional effects consistent with browning on energy expenditure in human primary adipocytes. Both the basal and succinate-stimulated (complex II) oxygen consumption rates of adipocytes treated with MOVA

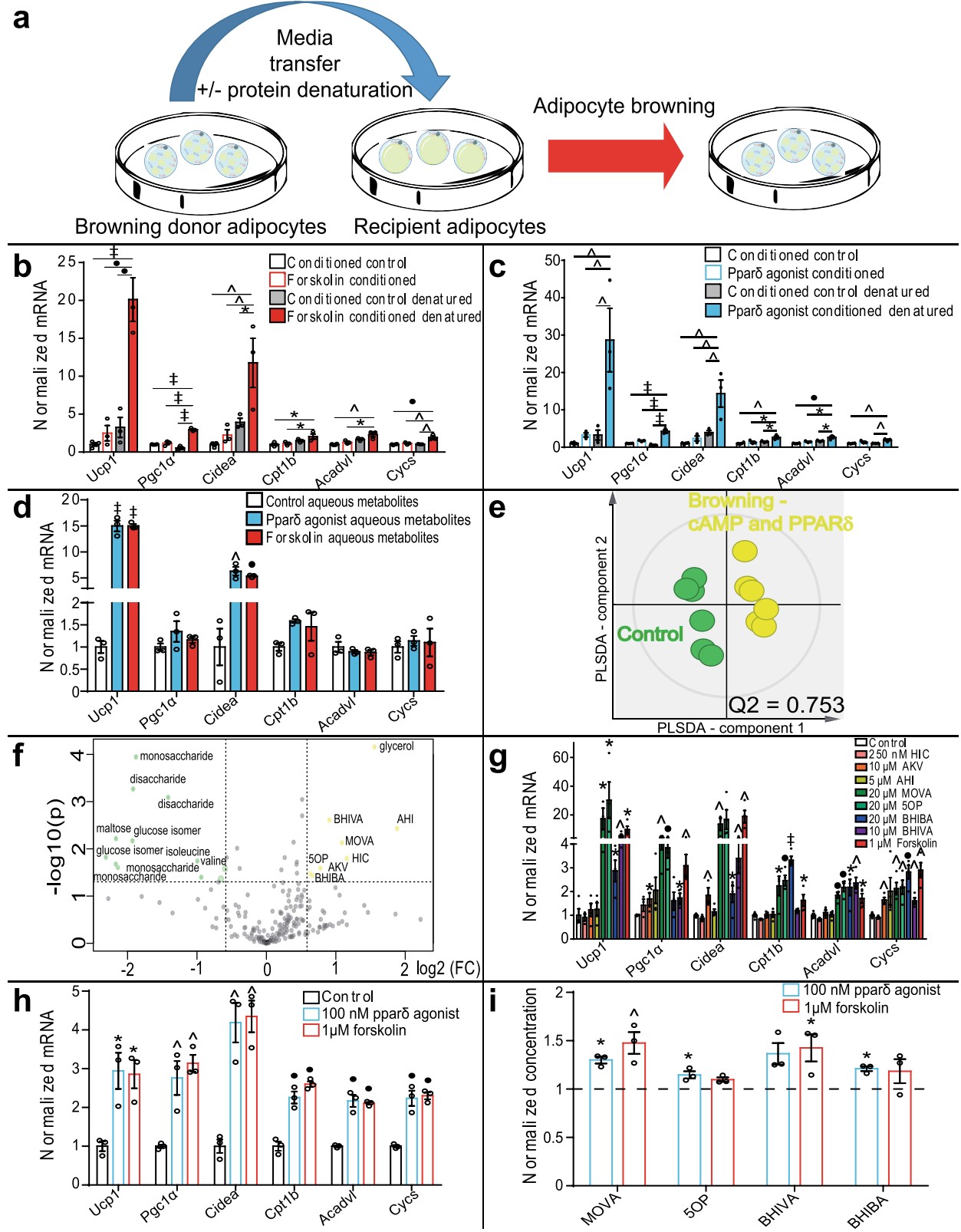

(20 μM), 5OP (20 μM), BHIBA (20 μM), and BHIVA (10 μM) were increased (Fig. 2d). Primary human adipocytes were treated with MOVA (20 μM), 5OP (20 μM), BHIBA (20 μM), and BHIVA (10 μM) and incubated in serum-free media containing U-$^{13}$C-palmitate to monitor adipocyte fatty acid β-oxidation. The labeled palmitate is catabolized via β-oxidation, releasing labeled acetyl CoA, which enters the TCA cycle (Supplementary Fig. 2e).

GC-MS analysis identified increased relative enrichment of downstream TCA cycle metabolites citrate, fumarate, and malate in MOVA, 5OP, and BHIBA-treated adipocytes (Fig. 2e–g), confirming that fatty acid β-oxidation is increased in these cells. The uptake of glucose and fatty acid into human primary adipocytes treated with the metabolites was measured with the fluorescent glucose analog 6-NBDG or the fluorescent fatty acid

**Fig. 1 Browning adipocytes secrete metabolites, which induce brown-adipocyte-associated gene expression in primary adipocytes. a** Brown-adipocyte-associated gene expression by murine primary adipocytes after exposure to conditioned media (±protein denaturation) from adipocytes induced to brown through cAMP (forskolin 1 μM; red) or peroxisome proliferator-activated receptor δ (PPARδ; GW0742 100 nM; light blue). **b, c** Conditioned media from browning adipocytes increases brown-adipocyte-associated gene expression in primary adipocytes. Denaturing the media protein content enhanced gene expression (control $n = 4$; Forskolin-conditioned, PPARδ agonist-conditioned, control denatured, Forskolin-conditioned denatured, PPARδ-conditioned denatured, $n = 3$; One-way ANOVA Tukey's post hoc; Control vs. Forskolin-conditioned denatured $Ucp1$ $P < 0.0001$, $Pgc1\alpha$ $P < 0.0001$, $Cidea$ $P = 0.003$, $Cpt1b$ $P = 0.019$, $Acadvl$ $P = 0.0012$, $Cycs$ $P = 0.0006$; Forskolin- vs. Forskolin-conditioned denatured $Ucp1$ $P = 0.0001$, $Pgc1\alpha$ $P < 0.0001$, $Cidea$ $P = 0.0095$, $Cpt1b$ $P = 0.047$, $Acadvl$ $P = 0.01$, $Cycs$ $P = 0.003$; conditioned control denatured vs. Forskolin-conditioned denatured $Ucp1$ $P = 0.002$, $Pgc1\alpha$ $P < 0.0001$, $Cidea$ $P = 0.029$, $Cycs$ $P = 0.0013$; Conditioned control vs. PPARδ agonist-conditioned denatured $Ucp1$ $P = 0.003$, $Pgc1\alpha$ $P < 0.0001$, $Cidea$ $P = 0.0013$, $Cpt1b$ $P = 0.0029$, $Acadvl$ $P = 0.0003$, $Cycs$ $P = 0.0049$; PPARδ agonist-conditioned vs. PPARδ agonist-conditioned denatured $Ucp1$ $P = 0.0075$, $Pgc1\alpha$ $P < 0.0001$, $Cidea$ $P = 0.0042$, $Cpt1b$ $P = 0.02$, $Acadvl$ $P = 0.0046$; conditioned control denatured vs. PPARδ agonist denatured $Ucp1$ $P = 0.0075$, $Pgc1\alpha$ $P < 0.0001$, $Cidea$ $P = 0.01$, $Cpt1b$ $P = 0.037$, $Acadvl$ $P = 0.017$, $Cycs$ $P = 0.0094$). **d** Reconstituted aqueous-soluble metabolites from browning adipocyte-conditioned media increases brown-adipocyte-associated gene expression in primary adipocytes ($n = 3$; One-way ANOVA Dunnett's post hoc; Forskolin aqueous metabolites $Ucp1$ $P < 0.0001$, $Cidea$ $P = 0.0023$; Pparδ agonist aqueous metabolites $Ucp1$ $P < 0.0001$, $Cidea$ $P = 0.0008$). **e** Metabolomic analysis of the browning adipocyte-conditioned media (yellow) separated from controls (green) in a partial least squares-discriminant analysis (PLS-DA) model ($n = 6$, $Q^2 = 0.753$). **f** Volcano plot analysis of metabolomic data identifies that the conditioned media from both browning models was enriched with α-hydroxyisocaproic acid (HIC), α-ketoisovaleric acid (AKV), α-hydroxyisovaleric acid (AHI), 3-methyl-2-oxovaleric acid (MOVA), 5-oxoproline (5OP), β-hydroxyisobutyric acid (BHIBA), and β-hydroxyisovaleric acid (BHIVA) ($n = 6$; fold-change threshold = 1.5, $P$ value threshold = 0.05). Metabolites enriched (yellow), metabolites depleted (green). **g** MOVA, 5OP, BHIBA, and BHIVA at physiological concentrations increased brown-adipocyte-associated gene expression in primary adipocytes. Forskolin treatment given as a positive control for browning. (control, AKV, AHI, MOVA, 5OP, BHIBA BHIVA, Forskolin $n = 4$; HIC $n = 3$; two-tailed $t$-test; AKV $Pgc1\alpha$ $P = 0.04$, $Cidea$ $P = 0.01$, $Cycs$ $P = 0.002$; MOVA $Ucp1$ $P = 0.02$, $Pgc1\alpha = 0.0015$, $Cidea$ $P = 0.006$, $Cpt1b$ $P = 0.02$, $Acadvl$ $P = 0.0009$, $Cycs$ $P = 0.005$; 5OP $Ucp1$ $P = 0.05$, $Pgc1\alpha$ $P = 0.0006$, $Cpt1b$ $P = 0.0007$, $Acadvl$ $P = 0.0008$, $Cycs$ $P = 0.007$; BHIBA $Ucp1$ $P = 0.012$, $Cidea$ $P = 0.04$, $Cpt1b$ $P = 0.000014$, $Acadvl$ $P = 0.016$, $Cycs$ $P = 0.0007$; BHIVA $Ucp1$ $P = 0.0068$, $Pgc1\alpha = 0.012$, $Cidea$ $P = 0.009$, $Acadvl$ $P = 0.001$, $Cycs$ $P = 0.005$; Forskolin $Ucp1$ $P = 0.011$, $Pgc1\alpha = 0.005$, $Cidea$ $P = 0.006$, $Cpt1b$ $P = 0.05$, $Acadvl$ $P = 0.017$, $Cycs$ $P = 0.001$). **h** Forskolin or a PPARδ agonist-induced thermogenic genes in mouse canonical primary brown adipocytes ($n = 3$; One-way ANOVA Dunnett's post hoc; Forskolin $UCP1$ $P = 0.014$, $PGC1\alpha$ $P = 0.003$, $CIDEA$ $P = 0.0016$, $CPT1b$ $P = 0.0002$, $ACADvl$ $P = 0.0003$, $CYCS$ $P = 0.0006$; PPARδ agonist $UCP1$ $P = 0.017$, $PGC1\alpha$ $P = 0.008$, $CIDEA$ $P = 0.002$, $CPT1b$ $P = 0.0006$, $ACADvl$ $P = 0.0002$, $CYCS$ $P = 0.0008$). **i** Conditioned media from primary brown adipocytes treated with either forskolin or PPARδ agonist was enriched with MOVA, 5OP, BHIBA, and BHIVA ($n = 3$; One-way ANOVA Dunnett's post hoc; Forskolin MOVA $P = 0.006$, BHIVA $P = 0.036$; PPARδ agonist MOVA $P = 0.046$, 5OP $P = 0.045$, BHIVA $P = 0.012$). (pink = HIC, orange = AKV, yellow = AHI, dark green = MOVA, light green = 5OP, dark blue = BHIBA, purple = BHIVA, red = forskolin) $*P \leq 0.05$, $\wedge P \leq 0.01$, $\bullet P \leq 0.001$, $\ddagger P \leq 0.0001$. Data are mean ± SEM with individual data points shown. Source data are provided as a Source Data file.

analog BODIPY-FA (Fig. 2h, i) (Supplementary Fig. 2f–m). Consistent with the browning response, the metabolites increased adipocyte glucose and fatty acid uptake.

We sought to further characterize the transcriptional program induced in adipocytes by the candidate metabolite signals, and to confirm that the effects on brown-adipocyte-associated gene expression are conserved in an independent in vitro model of human adipose tissue. A gene expression array of key adipocyte and brown-adipocyte-associated genes was used to probe immortalized human white preadipocytes isolated from neck fat and differentiated to mature adipocytes in the presence of MOVA (20 μM) (Supplementary Table 2), 5OP (20 μM) (Supplementary Table 3), BHIBA (20 μM) (Supplementary Table 4), or BHIVA (10 μM) (Supplementary Table 5). Confocal imaging of immortalized human adipocytes treated with the candidate metabokines identified MOVA, 5OP, and BHIBA significantly increased cellular UCP1 protein content (Fig. 2j, k). Functionally, basal and leak respiration are both increased in immortalized human adipocytes treated with the metabolites, partially due to increased electron flux seen as chemically uncoupled maximal respiration, and partially due to increased proton conductance seen as decreased coupling efficiency (Fig. 2l–o) (Supplementary Fig. 2n–q).

These data indicate that MOVA, 5OP, BHIBA, and, to a lesser extent, BHIVA induce gene and protein expression and a functional phenotype consistent with browning in two human adipocyte models.

**Transcriptional analysis and $^{13}$C-isotope substrate tracing reveal mechanisms of metabokine biosynthesis and secretion by browning adipocytes.** Next we examined mechanisms by which adipocyte browning may increase the concentrations of MOVA, BHIBA, BHIVA, and 5OP. BCAAs were depleted in the media of browning adipocytes (Fig. 1f). MOVA, BHIBA, and BHIVA are generated through the degradation of the BCAAs isoleucine, valine, and leucine, respectively. These pathways share multiple enzymes. 5OP is synthesized from glutamate. U-$^{13}$C-labeled isoleucine, valine, leucine, and glutamate were used to monitor stable isotope enrichment through the biosynthetic pathways and into extracellular accumulation of the candidate metabokines produced by human primary adipocytes treated with forskolin. Concomitantly, we performed RNA-Seq on human primary adipocytes treated with forskolin. Induction of the browning response increased both the intracellular and extracellular (culture media) $^{13}$C-enrichment of MOVA (Supplementary Fig. 3a–d), BHIBA (Supplementary Fig. 3e–j), BHIVA (Supplementary Fig. 3k–n), and 5OP (Supplementary Fig. 3o–u). The expression of the genes encoding BCAA catabolic enzymes, *branched-chain amino acid transaminase 2* (*BCAT2*), *branched-chain keto acid dehydrogenase E1 subunit beta* (*BCKDHB*), *acyl-CoA dehydrogenase short chain* (*ACADS*), *acyl-CoA dehydrogenase medium chain* (*ACADM*), *Enoyl-Coenzyme A, Hydratase/3-Hydroxyacyl Coenzyme A Dehydrogenase* (*EHHADH*), *hydroxyacyl-CoA dehydrogenase* (*HADHA*), and *Enoyl-CoA Hydratase, Short Chain 1* (*ECHS1*) was increased in forskolin-treated adipocytes (Supplementary Fig. 3a–n). The expression of the genes encoding 5OP biosynthetic enzymes *glutathione synthetase* (*GSS*), *γ-glutamyltransferase 7* (*GGT7*), and *γ-glutamylcyclotransferase* (*GGCT*) was also increased in browning adipocytes (Supplementary Fig. 3o–u).

These data identify that browning induces a transcriptional program upregulating expression of the metabokine biosynthetic enzymes and driving adipocyte synthesis and release of MOVA, 5OP, BHIBA, and BHIVA.

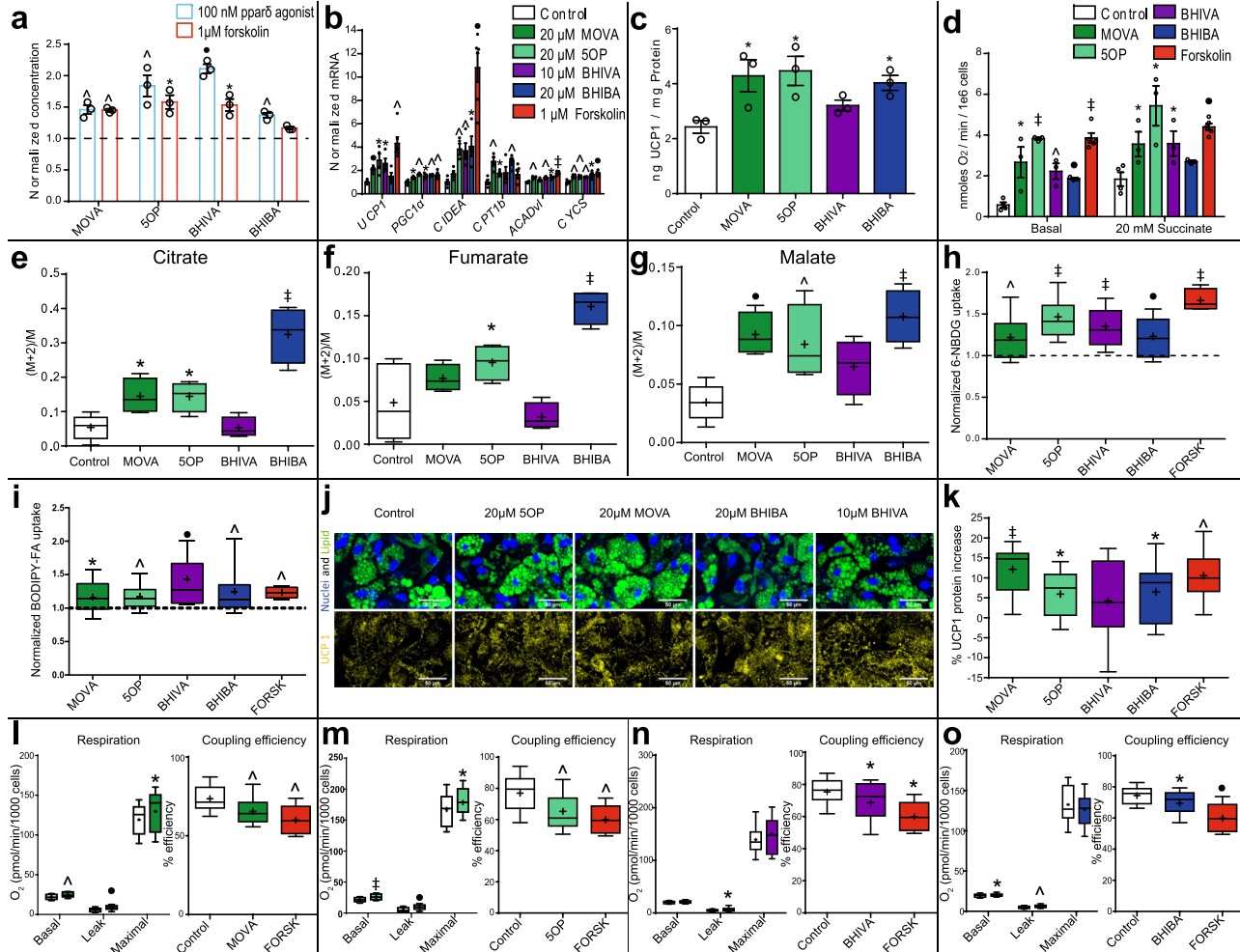

**Metabokines are exported from browning adipocytes via monocarboxylate transporters**. Next, we investigated the mechanisms through which the browning adipocytes export the metabolite signals. MOVA, 5OP, BHIBA, and BHIVA, structurally, share a common single carboxyl group. Our RNA-seq analysis identified that *monocarboxylate transporter 1 (MCT1)* expression was increased in human primary adipocytes treated with forskolin (*SLC16A1*, log fold-change = 0.37, $P < 0.05$, $n = 3$). MCT1 functions to both export and import monocarboxylates through the plasma membrane. 5OP, MOVA, and BHIVA are transported by MCT1[22–25]. MCTs also transport the BHIBA structurally related ketone body beta-hydroxybutyrate[22,26]. We used a pharmacological MCT inhibitor (MCTi, α-cyano-4-hydroxycinnamate) to determine the involvement of MCTs in browning-mediated secretion of the metabokines. Inhibition of MCTs abrogated forskolin-induced secretion of the metabokine signals, decreasing MOVA, 5OP, BHIBA, and BHIVA extracellular concentration whilst increasing their intracellular concentration (Supplementary Fig. 4a–d). To confirm MCT1 contributed to the browning-mediated secretion of the metabokines from adipocytes, we decreased *MCT1* expression in human adipocytes by 88% using siRNA (Fig. 3a). Knockdown of *MCT1* inhibited forskolin-induced export of the metabolites from adipocytes, again decreasing the metabolite extracellular concentration whilst increasing their intracellular concentration (Fig. 3b–e). Although not assessed with transport assays, these data suggest that MCTs are required for metabokine export from browning adipocytes.

**MOVA, 5OP, and BHIBA regulate metabolism in skeletal myocytes**. In murine models of both adipose tissue browning and BAT activity, fatty acid oxidation in skeletal muscle is increased[8–10]. We hypothesized that the metabolites secreted from browning adipocytes may contribute to the functional link between browning adipose tissue and muscle. We reconstituted this adipose tissue–muscle functional relationship in vitro. As previously described, conditioned serum-free media was collected from primary mouse adipocytes treated with forskolin (1 μM). Conditioned media was transferred to mouse C2C12 myotubes and induced expression of transcriptional regulators of metabolism (*Pparα*, *Pgc1α*), fatty acid β-oxidation genes including *Cpt1b*, and *Acadvl*, and mitochondrial genes *Cycs* and respiratory chain complex 1 component *NADH:Ubiquinone Oxidoreductase Core Subunit S1 (Ndufs1)* (Fig. 4a). The metabolites MOVA and 5OP induced expression of the metabolic gene panel in mouse myotubes (Fig. 4b).

The adipose tissue–muscle in vitro signaling model was translated to human primary cells. Conditioned media from browning human adipocytes induced expression of key fatty acid metabolism genes in human myocytes (Fig. 4c). The effect of MOVA and 5OP on metabolic, mitochondrial, and fatty acid oxidation gene expression was conserved in human primary skeletal myocytes and was dose responsive in the physiological low micromolar range (Fig. 4d) (Supplementary Fig. 5a–d). BHIBA was also observed to increase expression of *PPARα* and *CPT1b* in human primary skeletal myocytes. To confirm that transcriptional changes in human myocytes are accompanied by a dose-dependent change in functional phenotype, the oxygen

**Fig. 2 Browning human adipocytes secrete metabolites, which induce a brown-adipocyte-like functional phenotype. a** 3-methyl-2-oxovaleric acid (MOVA), 5-oxoproline (5OP), β-hydroxyisovaleric acid (BHIVA), and β-hydroxyisobutyric acid (BHIBA) are enriched in browning human adipocyte media ($n = 3$; One-way ANOVA Dunnett's post hoc; Forskolin MOVA $P = 0.0015$, 5OP $P = 0.034$, BHIVA $= 0.012$; PPARδ agonist MOVA $P = 0.0017$, 5OP $P = 0.007$, BHIVA $P = 0.0003$, BHIBA $P = 0.0017$). **b** MOVA, 5OP, BHIBA, and BHIVA induce brown-adipocyte-associated gene expression in human adipocytes. Forskolin treatment given as a positive control for browning (Control, MOVA, 5OP, BHIBA, and BHIVA $n = 4$; Forskolin $n = 6$; two-tailed $t$-test; MOVA Ucp1 $P = 0.0006$, Pgc1α $= 0.011$, Cpt1b $P = 0.005$, Acadvl $P = 0.003$, Cycs $P = 0.008$; 5OP Ucp1 $P = 0.019$, Pgc1α $= 0.0011$, Cidea $P = 0.002$, Cpt1b $P = 0.025$, Cycs $P = 0.0086$; BHIVA Ucp1 $P = 0.012$, Pgc1α $= 0.015$, Cidea $P = 0.009$, Acadvl $P = 0.0016$, Cycs $P = 0.0038$; BHIBA Pgc1α $P = 0.002$, Cidea $P = 0.014$, Cpt1b $P = 0.0024$, Acadvl $P = 0.01$, Cycs $P = 0.012$; Forskolin Ucp1 $P = 0.0014$, Pgc1α $= 0.009$, Cidea $P = 0.0002$, Acadvl $P < 0.0001$, Cycs $P = 0.0006$). **c** UCP1 protein concentration in human primary adipocytes treated with MOVA, 5OP, BHIVA, and BHIBA determined by ELISA ($n = 3$, One-way ANOVA Dunnett's post hoc; MOVA $P = 0.025$, 5OP $P = 0.015$, BHIBA $P = 0.05$). **d** Basal and stimulated (succinate 20 mmol/L) oxygen consumption increased in human adipocytes treated with MOVA, 5OP, BHIVA, and BHIBA, and Forskolin (provided for comparison) (Control $n = 4$, MOVA, 5OP, BHIVA, BHIBA $n = 3$, Forskolin $n = 5$; two-tailed $t$-test; Basal MOVA $P = 0.023$, 5OP $P < 0.0001$, BHIVA $P = 0.006$, BHIBA $P = 0.0005$, Forskolin $P < 0.0001$; 20-mM Succinate MOVA $P = 0.046$, 5OP $P = 0.011$, BHIVA $P = 0.044$, Forskolin $P = 0.00017$). **e–g** TCA cycle intermediates citrate, fumarate, and malate $^{13}C$-enrichment from $^{13}C$-palmitate metabolism in MOVA, 5OP, BHIVA, and BHIBA-treated human adipocytes. $M + n$, the isotope of M with an increased $m/z$ of $+n$ (Control $n = 10$, MOVA, 5OP, BHIVA, BHIBA $n = 4$; One-way ANOVA Dunnett's post hoc; Citrate MOVA $P = 0.013$, 5OP $P = 0.013$, BHIBA $P < 0.0001$; Fumarate 5OP $P = 0.02$, BHIBA $P < 0.0001$; Malate MOVA $P = 0.0003$, 5OP $P = 0.0016$, BHIBA $P < 0.0001$). **h** Glucose uptake in MOVA, 5OP, BHIVA, and BHIBA-treated human adipocytes. Forskolin provided for comparison (MOVA $n = 29$, 5OP $n = 28$, BHIVA $n = 30$, BHIBA $n = 30$; two-tailed $t$-test; MOVA $P = 0.0019$, 5OP $P < 0.0001$, BHIVA $P < 0.0001$, BHIBA $P = 0.0009$). **i** Fatty acid uptake in MOVA, 5OP, BHIBA, and BHIVA-treated human adipocytes. Forskolin provided for comparison (MOVA, BHIBA, BHIVA $n = 18$, 5OP $n = 32$; two-tailed t-test; MOVA $P = 0.02$, 5OP $P < 0.0012$, BHIVA $P = 0.00011$, BHIBA $P = 0.0029$, Forskolin $= 0.0041$). **j** Composite confocal images (top) of immortalized human adipocytes from neck fat treated with MOVA, 5OP, BHIBA, and BHIVA, stained for lipid (green), nuclei (blue), and UCP1 (yellow) (bottom) (representative images from control $= 9$, MOVA $= 11$, 5OP $= 9$, BHIBA $= 12$, BHIVA $= 12$; scale bars $= 50$ μm). **k** UCP1 in human adipocytes from neck fat treated with MOVA, 5OP, BHIBA, BHIVA, or Forskolin (percentage change to control) (MOVA $n = 11$, 5OP $n = 9$, BHIBA $n = 12$, BHIVA $n = 12$, Forskolin $n = 6$; two-tailed $t$-test; MOVA $P < 0.0001$, 5OP $P = 0.031$, BHIBA $P = 0.0395$, Forskolin $P = 0.0033$). Basal respiration, proton leak, chemically uncoupled maximal respiration, and coupling efficiency, assessed by the Seahorse XF platform Mito Stress assay, in human primary adipocytes isolated from neck fat and treated with MOVA **l** (control $n = 26$, MOVA $n = 29$; two-tailed $t$-test; Basal $P = 0.0063$, Leak $P = 0.00084$; maximal $P = 0.02$; One-way ANOVA with Dunnett's post hoc; coupling efficiency MOVA $P = 0.004$, Forskolin $P = 0.0066$) 5OP **m** (control $n = 29$, 5OP $n = 28$; two-tailed $t$-test; Basal $P < 0.0001$, Leak $P = 0.0002$, maximal $P = 0.02$ One-way ANOVA with Dunnett's post hoc; coupling efficiency 5OP $P = 0.0013$, Forskolin $P = 0.01$) BHIVA **n** (control $n = 28$, BHIVA $n = 30$; two-tailed $t$-test; Leak $P = 0.033$, One-way ANOVA with Dunnett's post hoc; coupling efficiency BHIVA $P = 0.037$, Forskolin $P = 0.02$) or BHIBA **o** (control $n = 29$, BHIBA $n = 29$; two-tailed $t$-test; Basal $P = 0.05$, Leak $P = 0.01$, One-way ANOVA with Dunnett's post hoc; coupling efficiency BHIBA $P = 0.03$, Forskolin $P = 0.0005$) and compared with forskolin ($n = 6$). Experiments were performed with 20 μM MOVA, 20 μM 5OP, 20 μM BHIBA, 10 μM BHIVA, and 1 μM Forskolin. ∗$P ≤ 0.05$, ^$P ≤ 0.01$, ●$P ≤ 0.001$, ‡$P ≤ 0.0001$. Light blue $=$ PPARδ agonist, red $=$ forskolin, dark green $=$ MOVA, light green $=$ 5OP, purple $=$ BHIVA, dark blue $=$ BHIBA. Data in bar charts are mean ± SEM with data points shown. Box and whisker plots show 25th to 75th percentile (box) min to max (whiskers), mean (+) and median (−). Source data are provided as a Source Data file.

consumption rates of primary myocytes treated with MOVA (5 and 20 μM), 5OP (5 and 20 μM), BHIBA (5 and 20 μM), and BHIVA (2.5 and 10 μM) were measured. Basal respiration rates of the myocytes were increased by MOVA (Supplementary Fig. 5e), 5OP (Supplementary Fig. 5f), and BHIBA (Supplementary Fig. 5g), but not BHIVA (Supplementary Fig. 5h). MOVA and 5OP induced the greatest increase in fatty acid oxidation gene expression in both mouse and human primary myocytes. These metabolites were selected for characterization in primary myocytes using a substrate-inhibitor high-resolution respirometry protocol. MOVA and 5OP increased respiratory capacity in permeabilized human myocytes supported by substrates for fatty acid β-oxidation (octanoyl-carnitine/malate/ADP) (Fig. 4e, f). 5OP also increased chemically uncoupled maximal substrate oxidation (CCCP) in myocytes (Fig. 4f). Murine models of adipose browning and thermogenesis activate fatty acid and glucose catabolism in skeletal muscle[8,9,15]. Therefore we investigated the effect of the metabokines on uptake of both glucose and fatty acid into human primary myocytes using the fluorescent glucose analog 6-NBDG and the fluorescent fatty acid analog BODIPY-FA, respectively (Fig. 4g, h) (Supplementary Fig. 5i–n).

These analyses identify that MOVA, 5OP, and, to a lesser extent, BHIBA regulate both murine and human skeletal myocyte metabolism consistent with an adipose–muscle metabolic signaling axis.

**Metabolite signals are enriched by cold conditioning and depleted by obesity in vivo.** To determine if MOVA, 5OP, BHIBA, and BHIVA function as brown and beige adipocyte metabokines in vivo we examined their concentrations in the BAT, subcutaneous inguinal WAT, and blood plasma of physiological models of increased and decreased adipose thermogenic function. We examined mice housed at thermoneutrality, room temperature, and under thermogenic conditions with cold exposure at 8 °C for a period of 1 week and 1 month. As expected, cold exposure robustly induced a thermogenic phenotype in mouse BAT and subcutaneous WAT gene expression, Ucp1 protein expression, and adipocyte morphology (Supplementary Fig. 6a–f). The concentrations of MOVA, 5OP, BHIBA, and BHIVA were increased in the BAT and subcutaneous WAT of cold-challenged mice (Fig. 5a, b). Consistent with the increase in the metabolite concentrations in the tissues, the expression of the MOVA, BHIBA, and BHIVA biosynthetic genes (Bcat2, Bckdhb, Acads, Acadm, Ehhadh, Hadha, and Echs1) and the 5OP biosynthetic genes (Gss, Ggct) were increased in the BAT (Fig. 5c) and subcutaneous WAT (Fig. 5d) of cold-challenged mice. The expression of the monocarboxylate transporter, Mct1, was also induced by cold challenge in the BAT (Fig. 5c) and subcutaneous WAT (Fig. 5d) of mice. In line with their potential as secreted brown and beige adipocyte paracrine and endocrine metabokines, plasma concentrations of the metabolites were also increased in mice housed in a cold environment (Fig. 5e).

Brown and beige adipose tissue is lost during the so called whitening effect associated with obesity, leading BAT to morphologically and metabolically resemble WAT (22). We determined if the metabolites MOVA, 5OP, BHIBA, and BHIVA are decreased by diet-induced obesity. Diet-induced obese mice,

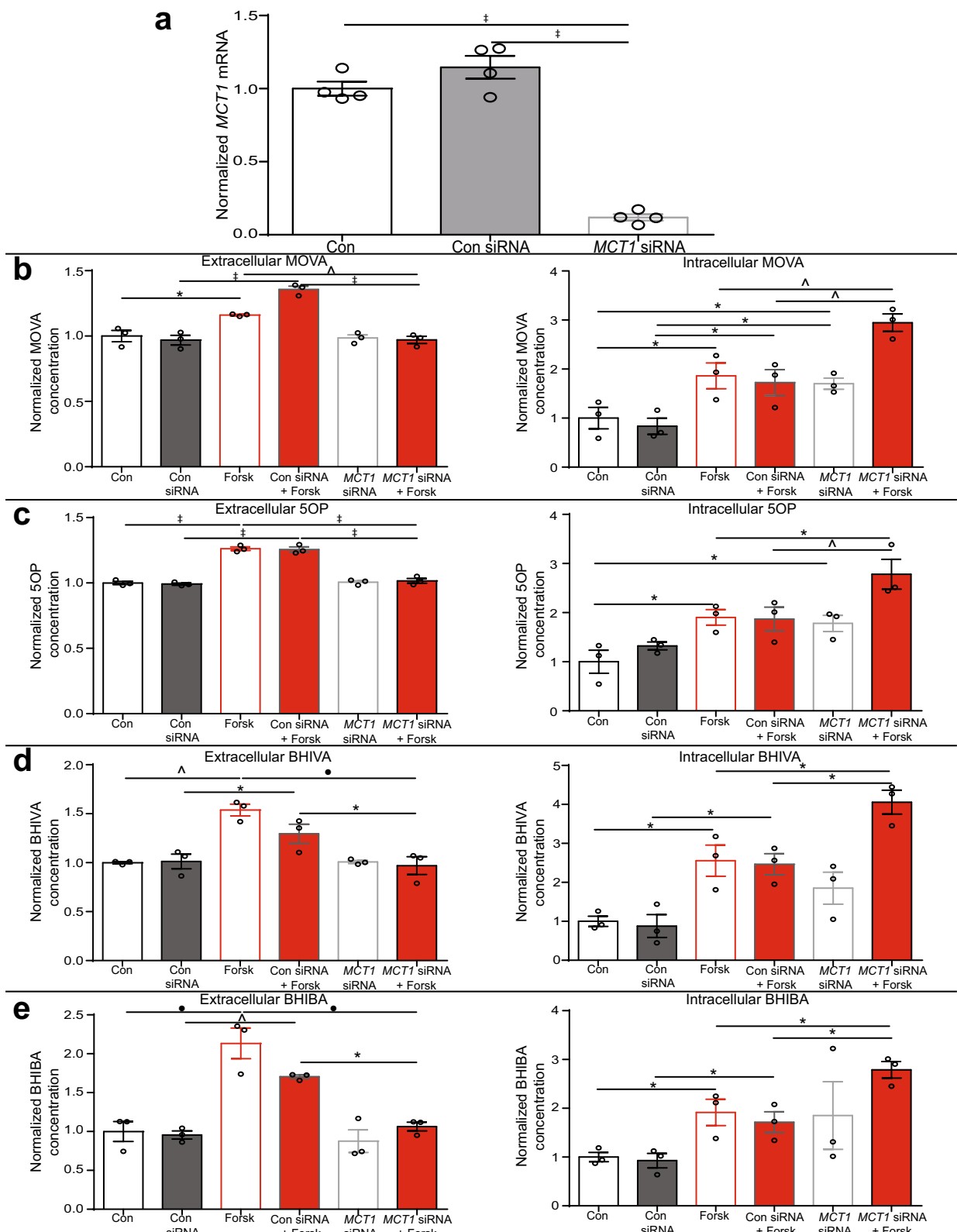

fed a 60% fat diet for 17 weeks, had greater body weight and impaired glucose tolerance compared to matched chow-fed controls (Supplementary Fig. 6g, h). Obese mice exhibited markers of whitening within their intrascapular BAT, with decreased expression of thermogenic genes (Supplementary Fig. 6i) and a white-adipocyte-like morphology (Supplementary Fig. 6j). In agreement with these observations, the BAT concentrations of MOVA, 5OP, BHIBA, and BHIVA were decreased by diet-induced obesity (Supplementary Fig. 6k).

Therefore the metabokine signals are modulated in adipose depots and systemically in in vivo physiological models of altered thermogenic function.

**Fig. 3 Export of metabolite signals from browning adipocytes is mediated by monocarboxylate transporter 1. a** The expression of monocarboxylate transporter 1 (MCT1) in primary human adipocytes treated with a scrambled control siRNA (con siRNA) or an siRNA against MCT1 (MCT1 siRNA) ($n = 4$; Con $P < 0.0001$, Con siRNA $P < 0.0001$). The concentration of **b** 3-methyl-2-oxovaleric acid (MOVA), **c** 5-oxoproline (5OP), **d** β-hydroxyisovaleric acid (BHIVA), and **e** β-hydroxyisobutyric acid (BHIBA) measured by liquid chromatography–mass spectrometry in adipocytes (intracellular) and the media (extracellular) of cells treated with forskolin (1 μM) (Forsk), con siRNA, MCT1 siRNA, con siRNA and forskolin (con siRNA + Forsk), or MCT1 siRNA and forskolin (MCT1 siRNA + Forsk) ($n = 3$) (Extracellular; Con vs. Forsk; MOVA $P = 0.02$, 5OP $P < 0.0001$, BHIVA $P = 0.0012$, BHIBA $P = 0.0002$; Con siRNA vs. Con SiRNA + Forsk MOVA $P < 0.0001$, 5OP $P < 0.0001$, BHIVA $P = 0.0122$, BHIBA $P = 0.0073$; Forsk vs. MCT1 siRNA + Forsk MOVA $P = 0.0061$, 5OP $P < 0.0001$, BHIVA $P = 0.0007$, BHIBA $P = 0.0004$; Con siRNA + Forsk vs. MCT1 siRNA + Forsk MOVA $P < 0.0001$, 5OP $P < 0.0001$, BHIVA $P = 0.047$, BHIBA $P = 0.022$) (Intracellular; Con vs. Forsk MOVA $P = 0.0123$, 5OP $P = 0.0103$, BHIVA $P = 0.04$ BHIBA $P = 0.032$; Con vs. MCT1 siRNA MOVA $P = 0.034$, 5OP $P = 0.022$; Con siRNA vs. Con siRNA + Forsk MOVA $P = 0.0101$, BHIVA $P = 0.037$, BHIBA $P = 0.039$; Con siRNA vs. MCT1 siRNA MOVA $P = 0.012$; Forsk vs. MCT siRNA + Forsk MOVA $P = 0.003$, 5OP $P = 0.012$, BHIVA $P = 0.05$, BHIBA $P = 0.05$; Con siRNA + Forsk vs. MCT1 siRNA + Forsk MOVA $P = 0.0013$, 5OP $P = 0.0097$, BHIVA $P = 0.036$, BHIBA $P = 0.041$). One-way ANOVA with Tukey's post hoc ∗$P \le 0.05$, ^$P \le 0.01$, ●$P \le 0.001$, ‡$P \le 0.0001$. Data are mean ± SEM with individual data points shown. Source data are provided as a Source Data file.

**Concentrations of the adipokine-like metabolites in adipose tissue and plasma are inversely correlated with body mass index in humans**. We investigated the association between genetic variants in the genes encoding the metabokine biosynthetic enzymes and body mass index (BMI) in a large-scale Genome Wide Association Study (GWAS) database in Genetic Investigation of ANthropometric Traits and UK Biobank Meta-analysis[27], included in the 795,640 subjects in the Type 2 Diabetes Knowledge Portal (http://www.type2diabetesgenetics.org/). We found that common noncoding variants in the MOVA, BHIBA, and BHIVA biosynthetic genes (*BCAT2*, *BCKDHB*, *ACADS*, and *HADHA*), the 5OP biosynthetic genes (*GSS*, *GGCT1*) and the gene for *MCT1* were significantly associated with BMI (Supplementary Table 6). The most significant variants in each gene for BMI were: *BCKDHB* rs13220420, $P = 0.00000750$; *BCAT2* rs73587808, $P = 0.000488$; *ACADS* rs12369156, $P = 0.000131$; *HADHA* rs559393527, $P = 0.0000341$; *GSS* rs2236270, $P = 3.60e$ $-8$; *GGCT* rs549124813, $P = 0.0000875$ and *MCT1/SLC16A1* rs186286251, $P = 0.000471$).

We then examined the association of subcutaneous WAT MOVA, 5OP, BHIVA, and BHIBA concentrations with BMI in human volunteers (Supplementary Table 7). The WAT concentration of MOVA, 5OP, and BHIVA were significantly inversely correlated with BMI (Fig. 6a–d). Plasma concentrations of the metabolite adipokine-like signals were also inversely correlated with BMI (Supplementary Fig. 7a–d).

The association of metabokine concentrations with beige adipose tissue in humans was also interrogated. RNA was isolated from the adipose tissue of volunteers and the expression of *UCP1* and *CPT1b* measured using RT-qPCR. Associations between the adipose tissue metabolite concentrations and the expression of *UCP1* (Fig. 6e–h) and *CPT1b* (Fig. 6i–l) were analyzed. Concentrations of MOVA, 5OP, and BHIVA were significantly correlated with tissue expression of *UCP1* and *CPT1b*.

These data suggest that the metabokines are functionally associated with human physiology and may influence body mass phenotypes.

**The metabokines MOVA, 5OP, and BHIBA increase systemic energy expenditure and regulate the adipose tissue and skeletal muscle metabolic phenotype in vivo**. Next, we investigated the effect of MOVA, 5OP, BHIBA, and BHIVA on the in vivo metabolic phenotype of mice. Six-week-old mice fed standard chow were either treated with MOVA (100 mg/kg/day), 5OP (100 mg/kg/day), BHIBA (150 mg/kg/day), or BHIVA (125 mg/kg/day) in drinking water for 17 weeks (based on preliminary dose escalation studies) or remained untreated (control mice). Treatment increased the plasma concentrations of the metabolites in the mice within the low micromolar physiological range.

(Supplementary Fig. 8a–d). Water intake was not different between groups (Supplementary Fig. 8e). Weight gain of 5OP- and MOVA-treated mice was decreased compared with controls (Supplementary Fig. 8f). Analysis with metabolic cages indicated BHIBA, MOVA, and 5OP increased energy expenditure (Supplementary Fig. 8g–j) and oxygen consumption (Supplementary Fig. 8k–n) independent of body mass (as determined by ANCOVA). Metabolite treatment did not affect the activity of the mice (Supplementary Fig. 8o). Food intake was increased in the 5OP and BHIBA-treated groups, which likely underpin the lack of difference in weight between BHIBA-treated mice and control (Supplementary Fig. 8p). BHIVA had no effect on the metabolic parameters independent of body mass.

MOVA, 5OP, and BHIBA increased systemic energy expenditure in mice. We examined the expression of thermogenic and mitochondrial metabolism genes in BAT, subcutaneous inguinal WAT, and skeletal muscle of the metabokine-treated mice (Supplementary Fig. 8q–t). Metabolite treatment also increased citrate synthase activity, a marker of mitochondrial density and TCA cycle flux, in the BAT, inguinal WAT, and muscle of metabokine-treated mice (Supplementary Fig. 8u–x).

**MOVA, 5OP, and BHIBA decrease weight gain, increase systemic energy expenditure, and regulate glucose homeostasis in a mouse model of obesity and diabetes**. The candidate metabokines 5OP, MOVA, and BHIBA increased energy expenditure and markers of oxidative metabolism in muscle and thermogenesis in adipose tissue in mice. We investigated the effect of MOVA, 5OP, and BHIBA on the metabolic phenotype in a high-fat feeding mouse model of obesity and T2DM. Six-week-old mice were treated with the metabokines (MOVA 100 mg/kg/day, 5OP 100 mg/kg/day, and BHIBA 150 mg/kg/day) in drinking water for 17 weeks while fed a 60% fat diet. Plasma concentrations of the metabolites were significantly increased in treated mice (Supplementary Fig. 9a–c). MOVA, 5OP, and BHIBA significantly reduced weight gain in fat-fed mice (Fig. 7a–c). Adiposity of the MOVA- and 5OP-treated mice was observed to be reduced by 17.1% and 19.4%, respectively, using computed tomography (CT) (Fig. 7d). Consistent with the effect on adiposity and body weight, analysis with metabolic cages indicated that whole-body energy expenditure (Fig. 7e–g) and oxygen consumption (Supplementary Fig. 9d–f) were increased in the MOVA-, 5OP-, and BHIBA-treated high-fat-fed mice, independent of body mass (as determined by ANCOVA). There was no significant difference in activity, food intake, or water intake (Supplementary Fig. 9g–i).

Next, the mice were challenged with an insulin tolerance test (Fig. 7h–j) (Supplementary Fig. 9j) and intraperitoneal glucose tolerance test (IPGTT) (Supplementary Fig. 9k–n). 5OP and BHIBA significantly improved both the insulin sensitivity and the

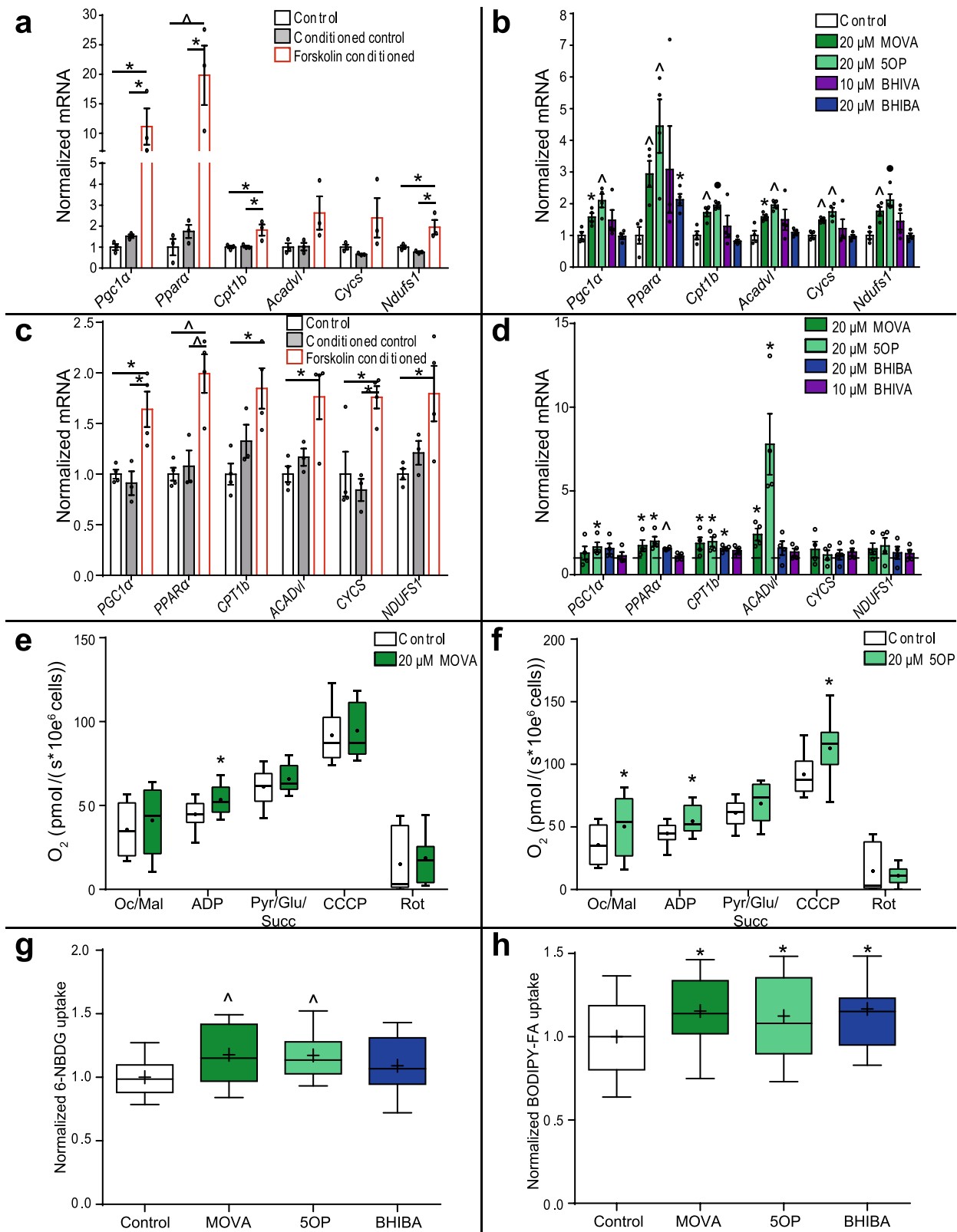

glucose tolerance in the mice. MOVA treatment demonstrated a mild but significant improvement in insulin sensitivity.

We then examined markers of thermogenesis and mitochondrial metabolism in BAT and subcutaneous WAT of the MOVA-, 5OP-, and BHIBA-treated mice. The activity of citrate synthase was significantly increased in the BAT of mice following MOVA, 5OP, and BHIBA treatment, suggesting increased mitochondrial

biogenesis (Fig. 7k). Consistent with these data, IHC analysis of the BAT of MOVA-, 5OP-, and BHIBA-treated mice indicated increased concentrations of Ucp1 (Supplementary Fig. 9o), which were confirmed by ELISA (Fig. 7l). Pgc1α protein concentration was also increased in the BAT of BHIBA-treated mice (Fig. 7l). Citrate synthase activity was increased in subcutaneous WAT of mice following 5OP and BHIBA treatment (Fig. 7m).

**Fig. 4 MOVA, 5OP, and BHIBA regulate metabolism in mouse and human skeletal myocytes. a** Conditioned media from murine primary adipocytes induced to brown via a cAMP-mediated mechanism increases expression of mitochondrial and metabolic genes in C2C12 myocytes ($n = 3$; One-way ANOVA with Tukey's post hoc; Control vs. Forskolin-conditioned, $Pgc1\alpha$ $P = 0.016$, $Ppar\alpha$ $P = 0.0092$, $Cpt1b$ $P = 0.027$, $Ndufs1$ $P = 0.041$; Conditioned Control vs. Forskolin-conditioned, $Pgc1\alpha$ $P = 0.02$, $Ppar\alpha$ $P = 0.011$, $Cpt1b$ $P = 0.031$, $Ndufs1$ $P = 0.016$). **b** The expression of the mitochondrial and metabolic gene panel in C2C12 myocytes treated with 3-methyl-2-oxovaleric acid (MOVA), 5-oxoproline (5OP), β-hydroxyisovaleric acid (BHIVA), and β-hydroxyisobutyric acid (BHIBA) ($n = 4$; two-tailed $t$-test; $Pgc1\alpha$ MOVA $P = 0.015$, 5OP $P = 0.004$; $Ppar\alpha$ MOVA $P = 0.008$, 5OP $P = 0.008$, BHIBA $P = 0.013$; $Cpt1b$ MOVA $P = 0.006$, 5OP $P = 0.0005$; $Acadvl$ MOVA $P = 0.011$, 5OP $P = 0.0013$; $Cycs$ MOVA $P = 0.0012$, 5OP $P = 0.002$; $Ndufs1$ MOVA $P = 0.006$, 5OP $P = 0.002$). **c** Conditioned media from browning human primary adipocytes increases expression of mitochondrial and metabolic genes in human primary skeletal myocytes (Control, Forskolin-conditioned $n = 4$, Control conditioned $n = 3$; One-way ANOVA with Tukey's post hoc; Control vs. Forskolin-conditioned, $Pgc1\alpha$ $P = 0.014$, $Ppar\alpha$ $P = 0.0027$, $Cpt1b$ $P = 0.012$, $Acadvl$ $P = 0.016$, $Cycs$ $P = 0.024$, $Ndufs1$ $P = 0.032$; Conditioned Control vs. Forskolin-conditioned, $Pgc1\alpha$ $P = 0.011$, $Ppar\alpha$ $P = 0.007$, $Cycs$ $P = 0.014$). **d** The expression of the mitochondrial and metabolic gene panel in human primary skeletal myocytes treated with MOVA, 5OP, BHIVA, and BHIBA ($n = 4$; two-tailed $t$-test; $PGC1\alpha$ 5OP $P = 0.037$; $PPAR\alpha$ MOVA $P = 0.043$, 5OP $P = 0.033$, BHIBA $P = 0.0011$; $CPT1b$ MOVA $P = 0.034$, 5OP $P = 0.018$, BHIBA $P = 0.011$; $ACADvl$ MOVA $P = 0.037$, 5OP $P = 0.016$). High-resolution respirometry analysis of human primary myocyte respiration with octanoyl-carnitine/malate (Oc/Mal) followed by ADP, pyruvate, glutamate and succinate (Pyr/Glu/Succ), maximal chemically uncoupled substrate oxidation (carbonyl-cyanide m-chlorophenyl hydrazine; CCCP), and rotenone (Rot) following treatment with **e** MOVA (20 µM) ($n = 12$; two-tailed $t$-test; ADP $P = 0.027$) or **f** 5OP (20 µM) (Control $n = 12$, 5OP $n = 11$; two-tailed $t$-test; Oc/Mal $P = 0.038$, ADP $P = 0.024$, CCCP $P = 0.03$). **g** Glucose uptake (6-(N-(7-Nitrobenz-2-oxa-1,3-diazol-4-yl)amino)-6-Deoxyglucose; 6-NBDG) in MOVA (20 µM), 5OP (20 µM), and BHIBA (20 µM)-treated human skeletal myocytes (Control $n = 32$, MOVA $n = 31$, 5OP $n = 32$, BHIBA $n = 34$; One-way ANOVA with Dunnett's post hoc; MOVA $P = 0.007$, 5OP $P = 0.0085$). **h** Fatty acid (BODIPY-FA) uptake in MOVA (20 µM), 5OP (20 µM), and BHIBA (20 µM)-treated human skeletal myocytes (Control $n = 57$, MOVA $n = 43$, 5OP $n = 43$, BHIBA $n = 47$; One-way ANOVA with Dunnett's post hoc; MOVA $P = 0.027$, 5OP $P = 0.046$, BHIBA $P = 0.019$). Dark green = MOVA, light green = 5OP, purple = BHIVA, dark blue = BHIBA. $*P \leq 0.05$, $\char94 P \leq 0.01$, $\bullet P \leq 0.001$, $\ddagger P \leq 0.0001$. Data in bar charts are mean ± SEM with data points shown. Box and whisker plots show 25th to 75th percentile (box) min to max (whiskers), mean (+) and median (−). Source data are provided as a Source Data file.

IHC analysis of inguinal subcutaneous WAT from these mice indicated increased Ucp1 concentrations following 5OP and BHIBA treatment (Supplementary Fig. 9o), which were again confirmed by ELISA (Fig. 7n). 5OP and BHIBA also increased the concentration of Pgc1α protein in inguinal WAT (Fig. 7n), with 5OP, BHIBA, and MOVA all increasing Cpt1 concentrations (Fig. 7n). MOVA and 5OP decreased adipocyte hypertrophy, significantly reducing adipocyte size within the inguinal WAT depot, consistent with effects of the metabolites on weight gain (Supplementary Fig. 9p).

MOVA, 5OP, and BHIBA increased expression of mitochondrial and metabolic genes in skeletal myocytes in vitro and in vivo. Consequently we investigated markers of mitochondrial metabolism in the soleus muscle of the MOVA-, 5OP-, and BHIBA-treated murine model of obesity. Mitochondrial density was increased in skeletal muscle by all three metabolite signals (Fig. 7o). Protein concentrations of Pgc1α and Ndufs1 were significantly increased in the muscle of metabolite-treated mice (Fig. 7p).

Positron emission tomography/computed tomography (PET/CT) using the glucose analogue [18]F-fluorodeoxyglucose ([18]F-FDG) was used to determine the tissue-specific metabolic effects of MOVA (100 mg/kg/day), 5OP (100 mg/kg/day), and BHIBA (150 mg/kg/day) treatment in vivo in the mouse model of obesity and T2DM[28] (Fig. 7q). The metabolic activity of BAT was significantly increased in BHIBA- and MOVA-treated mice (Fig. 7r). Hind limb skeletal muscle metabolic activity was increased in BHIBA-, MOVA-, and 5OP-treated mice (Fig. 7s), with forelimb muscle metabolic activity significantly increased in MOVA- and 5OP-treated mice (Fig. 7t).

The candidate metabokines are concomitantly increased in the plasma by stimulation of thermogenesis. MOVA and 5OP produced the most robust and significant reduction in weight gain and adiposity in high-fat-fed mice. We examined whether these metabolites would have combinatorial anti-obesity and anti-diabetic effects on systemic metabolism. Six-week-old mice were treated with a combination of MOVA (100 mg/kg/day) and 5OP (100 mg/kg/day) in drinking water for 17 weeks and fed a 60% fat diet. The combination of metabolites additively reduced weight gain when compared to either 5OP or MOVA treatments alone

(Supplementary Fig. 10a). CT analysis identified that a combination of MOVA and 5OP reduced body fat by 24.6% in treated mice compared with controls (Supplementary Fig. 10b). Glucose tolerance was further improved by a combination of 5OP and MOVA treatment (Supplementary Fig. 10c). PET/CT analysis using [18]F-FDG indicated that mice treated with both MOVA and 5OP had enhanced glucose uptake into the hind limb skeletal muscle when compared to the singly administered treatments (Supplementary Fig. 10d, e).

Together, these data show that the metabokines increase energy expenditure, reduce weight gain, improve glucose homeostasis, and increase glucose and fatty acid catabolism in BAT, WAT, and skeletal muscle. The results of MOVA and 5OP combinatorial studies also suggest MOVA and 5OP function through disparate mechanisms and that the small molecule adipokine-like signals function in concert to mediate systemic metabolism and anti-obesity effects on release from brown/beige adipose tissue.

**MOVA and 5OP signal through cAMP–PKA–p38 MAPK and BHIBA via mTOR to regulate adipocyte and myocyte metabolic gene expression.** We determined whether the metabokines function extracellularly or intracellularly at the human adipocyte to induce expression of *UCP1*. MCTs function to both import and export monocarboxylate species[29]. Treatment of human adipocytes with MOVA, 5OP, or BHIBA increased the intracellular concentration of the metabokines; this effect was abrogated by co-treatment with the MCTi (α-cyano-4-hydroxycinnamate) (Fig. 8a–c). Concomitant treatment of primary adipocytes with the MCTi and 5OP impaired 5OP-induced *UCP1* expression. (Fig. 8d). Conversely, inhibition of MCT activity did not impair MOVA or BHIBA-mediated *UCP1* expression, with dual metabokine and MCTi treatment trending toward increased *UCP1* expression compared to metabokine treatment alone (Fig. 8e, f). We then examined whether the metabokines signaled via similar mechanisms in human primary skeletal myocytes. Treatment of human skeletal myocytes with the metabokines increased their intracellular concentration; this effect was impaired by co-treatment with the MCTi (Fig. 8g–i). Combined MCTi and 5OP treatment impaired 5OP-mediated *CPT1b* expression in

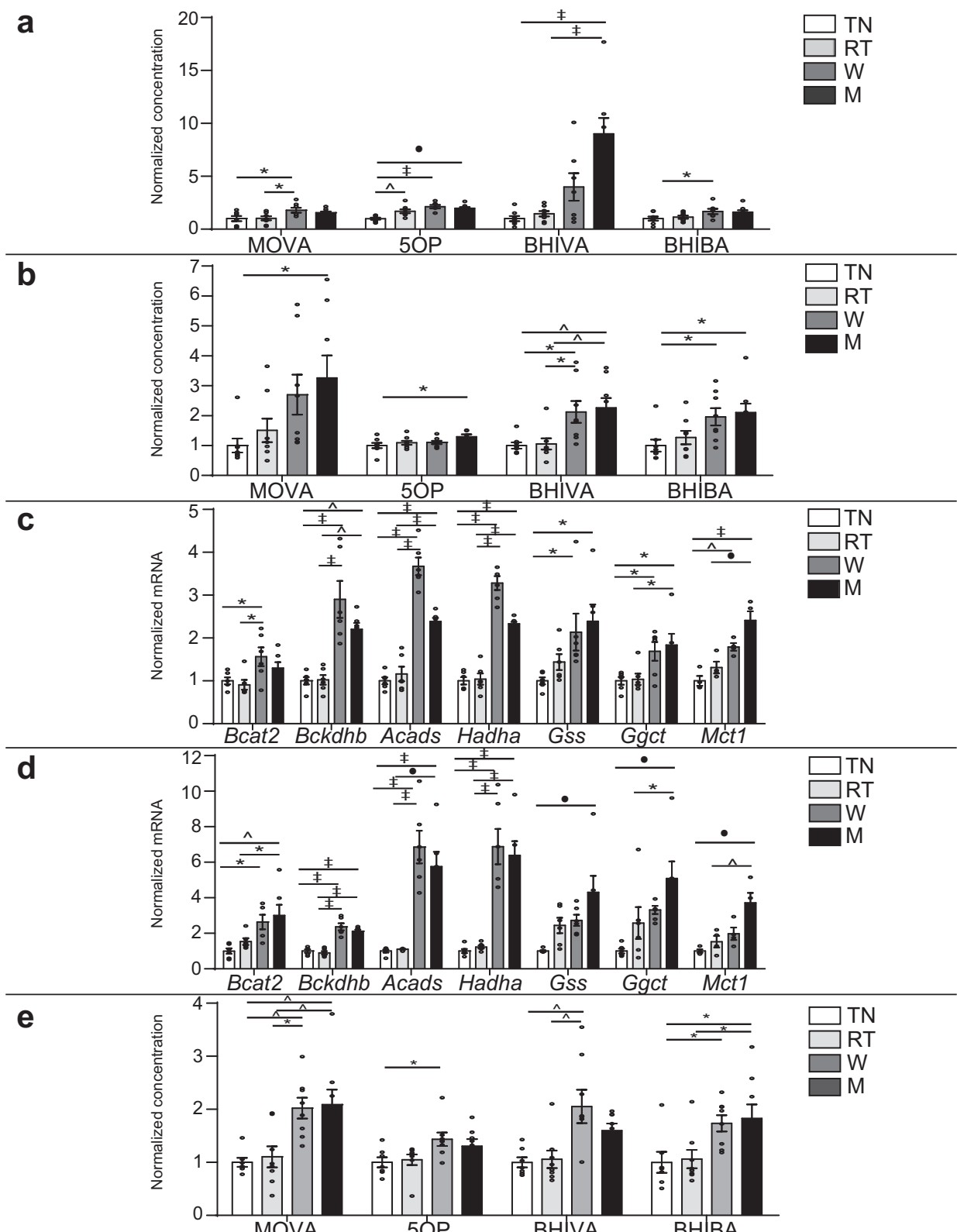

myocytes (Fig. 8j). The MOVA and BHIBA-induced expression of *CPT1b* was not impaired by MCTi (Fig. 8k, l). These data suggest that 5OP requires import into the cells to induce molecular signals leading to increased metabolic gene expression. Conversely these results indicate MOVA and BHIBA function through extracellular signal transduction and may require a receptor in the adipocyte and myocyte membrane.

Canonical activation of adipocyte thermogenesis through $\beta_3$-adrenergic signaling requires intracellular signal transduction by cAMP and downstream activation of protein kinase A (PKA)[1]. Using LC-MS, we measured the intracellular cAMP content in human adipocytes and myocytes treated with 5OP, MOVA, and BHIBA (Fig. 8m). The concentration of cAMP was unchanged in BHIBA-treated cells but increased in 5OP- and MOVA-treated

**Fig. 5 Cold exposure increases adipose tissue and circulating plasma concentrations of the metabokines.** GC-MS analysis of 3-methyl-2-oxovaleric acid (MOVA), 5-oxoproline (5OP), β-hydroxyisovaleric acid (BHIVA), and β-hydroxyisobutyric acid (BHIBA) concentration in the **a** interscapular brown adipose tissue (BAT) and **b** subcutaneous inguinal white adipose tissue (WAT) of mice housed at thermoneutrality (TN), room temperature (RT), 8 °C for 1 week (W), or 8 °C for 1 month (M) (BAT; TN $n = 8$, RT $n = 8$, W $n = 7$, M $n = 8$; One-way ANOVA with Dunnett's post hoc; MOVA TN vs. W $P = 0.029$, RT vs. W $P = 0.037$; 5OP TN vs. RT $P = 0.0046$, TN vs. W $P < 0.0001$, TN vs. M $P = 0.0001$; BHIVA TN vs. M $P < 0.0001$, RT vs. M $P < 0.0001$; BHIBA TN vs. W $P = 0.05$) (WAT $n = 8$; One-way ANOVA with Dunnett's post hoc; MOVA TN vs. M $P = 0.02$; 5OP TN vs. M $P = 0.026$; BHIVA TN vs. W $P = 0.016$, TN vs. M $P = 0.0067$, RT vs. W $P = 0.023$, RT vs. M $P = 0.0096$; BHIBA TN vs. W $P = 0.034$, TN vs. M $P = 0.013$). Expression of branched-chain amino acid catabolic (MOVA, BHIBA and BHIVA biosynthetic) enzymes *branched-chain amino acid transaminase 2* (*Bcat2*), *branched-chain keto acid dehydrogenase E1 subunit beta* (*Bckdhb*), *acyl-CoA dehydrogenase short chain* (*Acads*), *hydroxyacyl-CoA dehydrogenase* (*Hadha*), 5OP biosynthetic enzymes *glutathione synthetase* (*Gss*), *γ-glutamylcyclotransferase* (*Ggct*), and *monocarboxylate transporter 1* (*Mct1*) in the **c** interscapular BAT (*Bcat2*, TN vs. W $P = 0.035$, RT vs. W $P = 0.013$; *Bckdhb* TN vs. W $p < 0.0001$, TN vs. M $P = 0.005$, RT vs. W $P < 0.0001$, RT vs. M $P = 0.006$; *Acads* TN vs. W $P < 0.0001$, TN vs. M $P < 0.0001$, RT vs. W $P < 0.0001$, RT vs. M $P < 0.0001$; *Hadha* TN vs. W $P < 0.0001$, TN vs. M $P < 0.0001$, RT vs. W $P < 0.0001$, RT vs. M $P < 0.0001$; *Gss* TN vs. W $P = 0.04$, TN vs. M $P = 0.01$; *Ggct* TN vs. W $P = 0.045$, TN vs. M $P = 0.014$, RT vs. M $P = 0.018$; *Mct1* TN vs. W $P = 0.0062$, TN vs. M $P < 0.0001$, RT vs. M $P = 0.0005$) and **d** subcutaneous WAT (*Bcat2*, TN vs. W $P = 0.019$, TN vs. M $P = 0.0042$, RT vs. M $P = 0.037$; *Bckdhb* TN vs. W $P < 0.0001$, TN vs. M $P < 0.0001$, RT vs. W $P < 0.0001$, RT vs. M $P < 0.0001$; *Acads* TN vs. W $P < 0.0001$, TN vs. M $P < 0.0001$, RT vs. W $P < 0.0001$, RT vs. M $P = 0.0001$; *Hadha* TN vs. W $P < 0.0001$, TN vs. M $P < 0.0001$, RT vs. W $P < 0.0001$, RT vs. M $P < 0.0001$; *Gss* TN vs. M $P = 0.0008$; *Ggct* TN vs. M $P = 0.001$, RT vs. M $P = 0.04$; *Mct1* TN vs. M $P = 0.0006$, RT vs. M $P = 0.003$) of mice housed at thermoneutrality (TN), room temperature (RT), 8 °C for 1 week (W) or 8 °C for 1 month (M) ($n = 6$; One-way ANOVA with Dunnett's post hoc). **e** MOVA, 5OP, BHIVA, and BHIBA concentration in the blood plasma of mice housed at thermoneutrality (TN), room temperature (RT), 8 °C for 1 week (W) or 8 °C for 1 month (M) (TN $n = 7$, RT $n = 8$, W $n = 7$, M $n = 8$; One-way ANOVA with Dunnett's post hoc; MOVA TN vs. W $P = 0.006$, TN vs. M $P = 0.0035$, RT vs. W $P = 0.011$, RT vs. M $P = 0.0063$; 5OP TN vs. W $P = 0.042$; BHIVA TN vs. W $P = 0.0033$, RT vs. W $P = 0.0039$; BHIBA TN vs. W $P = 0.047$, TN vs. M $P = 0.023$, RT vs. M $P = 0.03$). $*P \leq 0.05$, $^\wedge P \leq 0.01$, $\bullet P \leq 0.001$, $\ddagger P \leq 0.0001$. Data in bar graphs are mean ± SEM with individual data points shown. Source data are provided as a Source Data file.

cells. We analyzed the cAMP content of BAT (Supplementary Fig. 11a), subcutaneous WAT (Supplementary Fig. 11b), and soleus skeletal muscle (Supplementary Fig. 11c) of 5OP-, MOVA-, and BHIBA-treated mice. The cAMP content was increased in BAT, subcutaneous WAT, and skeletal muscle of 5OP- and MOVA-treated mice. We then co-treated primary adipocytes with either MOVA or 5OP and the selective PKA inhibitor H89. Inhibition of PKA impaired MOVA and 5OP-induced expression of brown-adipocyte-associated genes in the adipocytes (Fig. 8n, o).

The downstream signaling pathways induced by MOVA, 5OP and BHIBA in skeletal muscle and adipose tissue were investigated using phosphokinase profiling of BAT, subcutaneous WAT, and soleus muscle from metabolite-treated mice. Treatment with MOVA increased phosphorylation of members of the p38 mitogen-activated protein kinase family (p38 MAPK) in BAT (Supplementary Fig. 11d), subcutaneous WAT (Supplementary Fig. 11e), and soleus (Supplementary Fig. 11f). In soleus, phosphorylation of the p38 MAPK substrate glycogen synthase kinase 3 beta (GSK-3β) was also increased by MOVA. In BAT, MOVA treatment increased phosphorylation of mitogen-activated protein kinase 3 located upstream of the p38 MAPKs in cellular signaling cascades. 5OP was also observed to increase phosphorylation of p38 MAPKs in BAT (Supplementary Fig. 11g), subcutaneous WAT (Supplementary Fig. 11h), and soleus (Supplementary Fig. 11i). These findings are consistent with the requirement for p38 MAPK in cAMP-mediated expression of *UCP1*[30]. BHIBA increased phosphorylation of both the mammalian target of rapamycin (mTOR) and mTOR's downstream substrate p70S6 kinase (p70S6K) in BAT, subcutaneous WAT, and soleus (Supplementary Fig. 11j–l) and increased the phospho mTOR/total mTOR ratio (Supplementary Fig. 11m, n). Inhibition of p38 MAPK signaling by a pan p38 MAPK inhibitor (500-nM BIRB 796) abrogated MOVA (Fig. 8p) and 5OP (Fig. 8q) induced metabolic gene expression (PPARα, CPT1b, ACADvl) in human primary skeletal myocytes. Inhibition of mTOR (temsirolimus 500 nM) in skeletal myocytes impaired BHIBA-induced metabolic gene expression (Fig. 8r). In human primary adipocytes, the inhibition of p38 MAPK signaling reduced MOVA and 5OP-induced brown-adipocyte-associated gene expression (Fig. 8s, t). Inhibition of mTOR signaling with temsirolimus reduced

BHIBA-induced expression of brown-adipocyte-associated genes in human primary adipocytes (Fig. 8u).

These data suggest that MOVA and 5OP function differentially via extracellular and intracellular mechanisms, respectively, to induce metabolic reprogramming in skeletal myocytes and adipocytes via a cAMP–PKA–p38 MAPK-mediated signaling pathway. BHIBA induces the thermogenic genes in adipocytes and mitochondrial gene expression in myocytes through extracellular activation of downstream mTOR signaling.

## Discussion

The canonical role of BAT, and to some extent beige adipose tissue, has long been regarded as to generate heat through uncoupled respiration. However, BAT and beige adipose tissue may have a more varied capacity to regulate systemic metabolism. The ability of WAT to function as an endocrine organ, releasing messengers known as adipokines that coordinate a systemic response to energetic status, feeding behaviors and inflammatory responses, amongst other physiological processes, is well established[31]. A similar endocrine function of BAT and beige adipose tissue remains poorly understood and characterized. However, evidence suggests the presence of endocrine and paracrine signals emanating from BAT and beige adipose tissue. Transplantation studies of both beige and brown fat in mice have demonstrated the capacity to induce weight loss in mouse models of obesity. Surprisingly, these studies have identified direct improvements in glucose homeostasis in skeletal muscle and activation of endogenous beige and brown fat[8,9,32]. In addition, the total loss of BAT has a more profound effect on metabolic status than the tissue-specific ablation of Ucp1. The ability of BAT to influence systemic energy balance is not solely reliant on non-shivering thermogenesis[11–14].

Recent discoveries have highlighted a number of peptidic, lipid, and miRNA brown adipokines[33], including the discovery by Kong and colleagues of a BAT-to-muscle signaling axis mediated through the protein myostatin[10]. We have previously highlighted a small molecule metabolite-mediated skeletal muscle to beige adipose tissue signaling axis[34]. Here we identify MOVA, 5OP, and BHIBA as a discrete set of brown and beige adipose metabokines. The effects of MOVA, 5OP, and BHIBA in vitro and in vivo have been summarized and compared in Supplementary

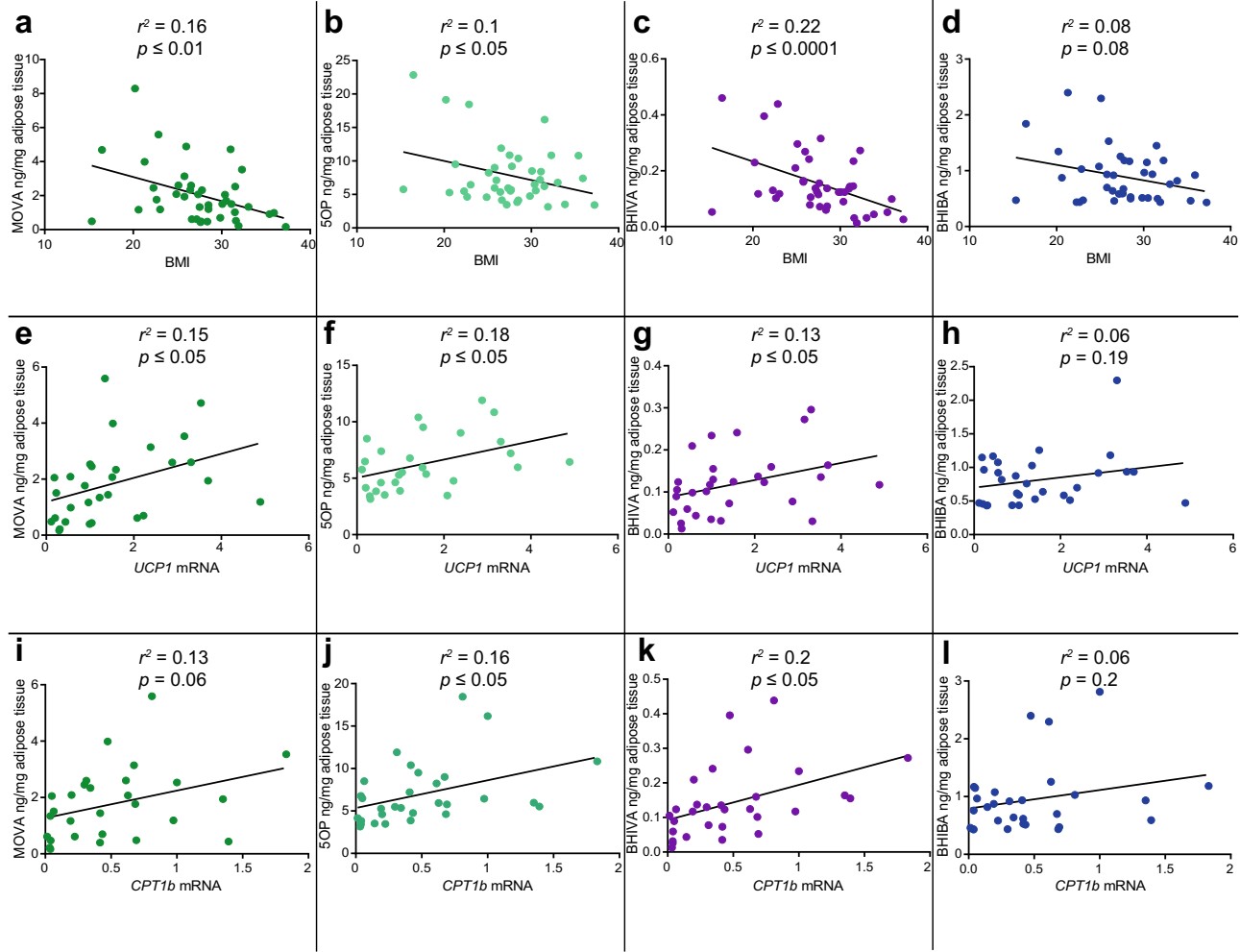

**Fig. 6 The adipokine-like metabolite adipose tissue concentrations are inversely correlated with BMI and positively correlated with brown-adipocyte-associated gene expression in humans. a–d** The inverse correlation of 3-methyl-2-oxovaleric acid (MOVA) ($n = 42$, $r^2 = 0.16$, $P = 0.0088$), 5-oxoproline (5OP) ($n = 42$, $r^2 = 0.1$, $P = 0.046$), β-hydroxyisovaleric acid (BHIVA) ($n = 42$, $r^2 = 0.345$, $P < 0.0001$), and β-hydroxyisobutyric acid (BHIBA) ($n = 41$, $r^2 = 0.08$, $P = 0.08$) subcutaneous adipose tissue concentration, measured by GC-MS, to BMI in human volunteers. **e–h** Correlation of MOVA ($n = 30$, $r^2 = 0.15$, $P = 0.032$), 5OP ($n = 30$, $r^2 = 0.18$, $P = 0.02$), BHIVA ($n = 30$, $r^2 = 0.13$, $P = 0.05$), and BHIBA ($n = 29$, $r^2 = 0.06$, $P = 0.19$) concentration with *uncoupling protein 1* (*UCP1*) gene expression in subcutaneous adipose tissue of human volunteers. **i–l** Correlation of MOVA ($n = 28$, $r^2 = 0.13$, $P = 0.06$), 5OP ($n = 30$, $r^2 = 0.164$, $P = 0.027$), BHIVA ($n = 30$, $r^2 = 0.2$, $P = 0.014$), and BHIBA ($n = 29$, $r^2 = 0.06$, $P = 0.2$) concentration with *carnitine palmitoyltransferase 1b* (*CPT1b*) gene expression in human adipose tissue. Dark green = MOVA, light green = 5OP, purple = BHIVA, dark blue = BHIBA. Analysis by Pearson correlation. Source data are provided as a Source Data file.

Table 8. These metabolites function in concert to mediate crosstalk between BAT, beige adipose tissue, and skeletal muscle by inducing expression of key mitochondrial genes and an oxidative phenotype in adipose and muscle, increasing whole-body energy expenditure, complementary to BATokine proteins and lipokines.

MOVA, BHIBA, and BHIVA are monocarboxylic acids generated via the catabolism of BCAAs. Catabolites of BCAAs are emerging as bioactive metabolites and have been implicated in endocrine signaling. Notably, the valine catabolite β-aminoisobutyric acid acts as an exercise-stimulated myokine, signaling to induce browning of WAT and hepatic fatty acid oxidation[34]. BHIBA is also a product of valine metabolism and signals between skeletal muscle and the endothelium in a PGC1α-dependent manner to increase fatty acid uptake[35]. Increased circulating plasma concentrations of BCAAs are a distinguishing feature of obesity and may predict T2DM onset[36–40]. Indeed, BCAAs can regulate protein synthesis and degradation, insulin secretion, and energy balance[41]. However, adipose tissue is capable of BCAA catabolism and can modulate circulating BCAA concentrations[42]. The rate of BCAA catabolism is particularly high in BAT and can regulate whole-body energy homeostasis[43–45]. BCAA catabolic enzymes are downregulated in adipose tissue both in obesity and in insulin resistance[44,46]. Here, we identify increased BCAA degradation to generate MOVA, BHIBA, and BHIVA in browning adipose tissue, a positive correlation between the subcutaneous adipose tissue concentration of these BCAA catabolites and brown-adipocyte-associated gene expression in humans, and an inverse correlation between the adipose tissue concentration of the metabolites and BMI. We also describe noncoding variants in the metabolite biosynthetic genes, which are significantly associated with BMI in large population GWAS meta-analysis. There is a loss of beige adipose tissue and BAT in the whitening effect associated with obesity[47]. In addition, the disruption of BCAA catabolism in BAT impairs thermogenesis and reduces tissue Ucp1[48,49]. Our work provides a link between the dysregulation of BCAA catabolism in adipose tissue, whitening of adipose depots, increased circulating BCAAs, and

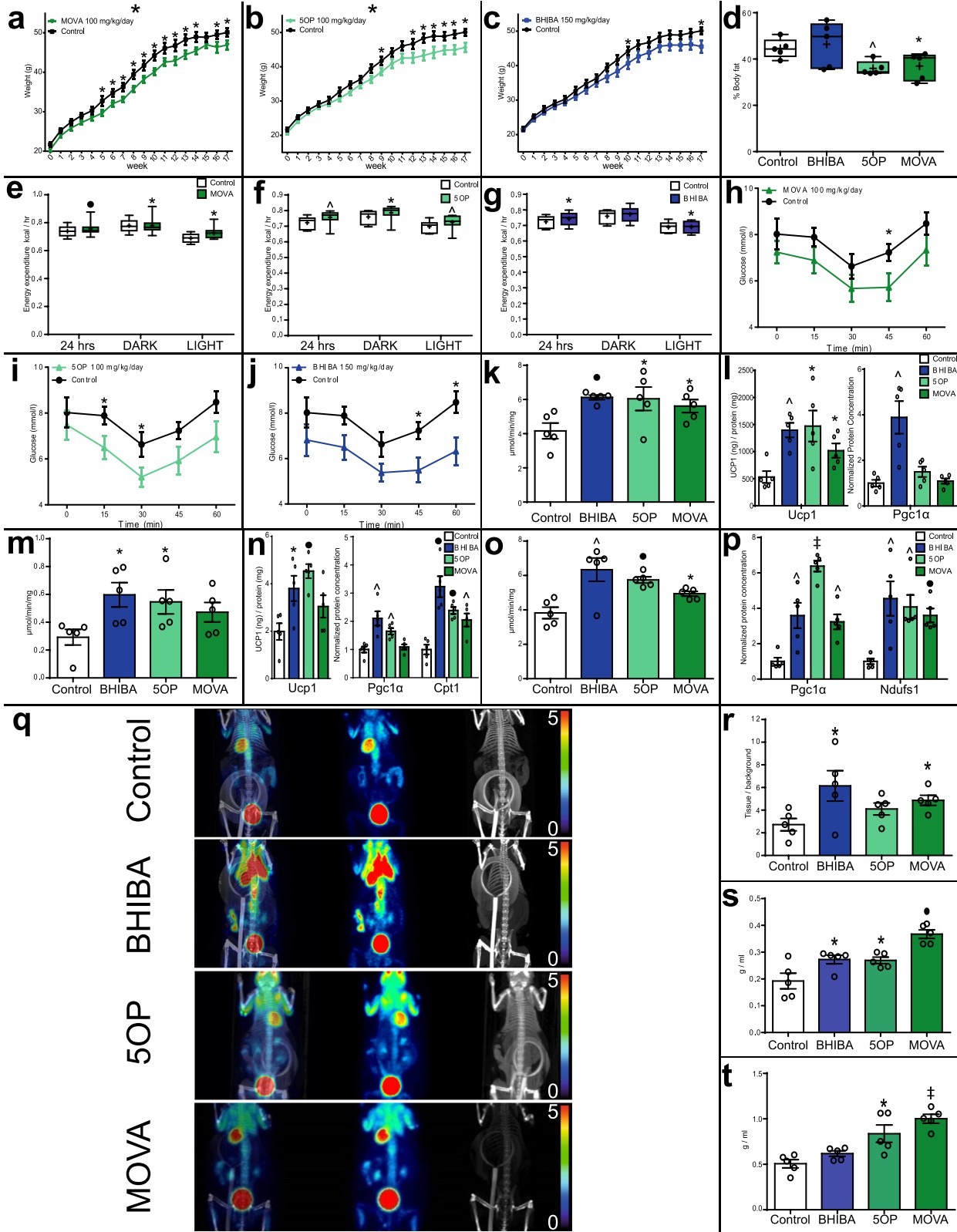

metabolic diseases including obesity. Metabolic risk associated with increased circulating BCAAs may, in part, be mediated by decreased biosynthesis and secretion of these brown and beige adipocyte metabokines and perturbation of the interorgan signaling axes they mediate.

5OP is a neglected amino acid, which links glutamine metabolism to glutathione biosynthesis. Glutathione is a potent

antioxidant and protects against adverse cellular redox states. Browning of adipose tissue has been postulated as an adaptive mechanism to alleviate redox pressure[50], with recent studies demonstrating that the expression of UCP1 is regulated by redox status including oxidative stress and antioxidants[51,52]. Thus, our identification of 5OP as a brown and beige metabokine may represent a mechanism through which beige/brown adipose can

**Fig. 7 MOVA, 5OP, and BHIBA decrease weight gain and regulate systemic energy metabolism in mice.** Weights of mice receiving **a** 100 mg/kg/day 3-methyl-2-oxovaleric acid (MOVA) ($n = 10$; Two-way ANOVA $P = 0.03$; two-tailed $t$-test, week 5 $P = 0.04$, week 6 $P = 0.046$, week 7 $P = 0.025$, week 8 $P = 0.01$, week 9 $P = 0.03$, week 10 $P = 0.01$, week 11 $P = 0.045$, week 12 $P = 0.046$, week 13 $P = 0.03$, week 14 $P = 0.03$, week 16 $P = 0.049$, week 17 $P = 0.02$) **b** 100 mg/kg/day 5-oxoproline (5OP) ($n = 10$; Two-way ANOVA $P = 0.05$; two-tailed $t$-test, week 8 $P = 0.049$, week 12 $P = 0.018$, week 13 $P = 0.02$, week 14 $P = 0.038$, week 15 $P = 0.036$, week 16 $P = 0.02$, week 17 $P = 0.02$), **c** 150 mg/kg/day β-hydroxyisobutyric acid (BHIBA) ($n = 10$; two-tailed $t$-test, week 10 $P = 0.05$, week 17 $P = 0.047$) compared to controls ($n = 9$). **d** Adiposity of MOVA-, 5OP-, and BHIBA-treated mice ($n = 5$; two-tailed $t$-test; MOVA $P = 0.006$, 5OP $P = 0.045$). Average hourly energy expenditure for 24-h period, 12-h dark phase (DARK) and 12-h light phase (LIGHT) of **e** MOVA ($n = 9$; ANCOVA with body mass as a covariate, 24 h $P < 0.001$, DARK $P = 0.017$, LIGHT $P = 0.011$) **f** 5OP ($n = 9$; ANCOVA with body mass as a covariate, 24 h $P = 0.0042$, DARK $P = 0.03$, LIGHT $P = 0.0044$) and **g** BHIBA ($n = 8$, ANCOVA with body mass as a covariate, 24 h $P = 0.013$, LIGHT $P = 0.03$) treated mice. Insulin tolerance tests of **h** MOVA (two-tailed $t$-test; 45 min $P = 0.049$), **i** 5OP (two-tailed $t$-test; 15 min $P = 0.049$, 30 min $P = 0.05$), and **j** BHIBA (two-tailed $t$-test, 45 min $P = 0.019$, 60 min $P = 0.014$) treated mice ($n = 10$, control $n = 9$). **k** Increased mitochondrial content in brown adipose tissue (BAT) of BHIBA-, 5OP-, and MOVA-treated mice ($n = 5$; two-tailed $t$-test, BHIBA $P = 0.004$, 5OP $P = 0.05$, MOVA $P = 0.04$). **l** Ucp1 and Pgc1α concentration of BAT from BHIBA-, 5OP-, and MOVA-treated mice ($n = 5$; two-tailed $t$-test; Ucp1 BHIBA $P = 0.001$, 5OP $P = 0.015$, MOVA $P = 0.02$; Pgc1α BHIBA $P = 0.004$). **m** Increased subcutaneous inguinal adipose tissue mitochondrial content in BHIBA and 5OP-treated mice ($n = 5$; two-tailed $t$-test; BHIBA $P = 0.019$, 5OP $P = 0.04$). **n** Ucp1, Pgc1α, and Cpt1 concentration of subcutaneous adipose tissue from BHIBA-, 5OP-, and MOVA-treated mice ($n = 5$; two-tailed $t$-test; Ucp1 BHIBA $P = 0.02$, 5OP $P = 0.005$; Pgc1α BHIBA $P = 0.004$, 5OP $P = 0.003$; Cpt1 BHIBA $P = 0.0006$, 5OP $P = 0.0002$, MOVA $P = 0.0065$). **o** BHIBA, 5OP, and MOVA increase soleus muscle mitochondrial content in mice ($n = 5$; two-tailed $t$-test; BHIBA $P = 0.01$, 5OP $P = 0.001$, MOVA $P = 0.015$). **p** BHIBA, 5OP, and MOVA increase Pgc1α and Ndufs1 in the soleus muscle of mice ($n = 5$; two-tailed $t$-test; Pgc1α BHIBA $P = 0.0085$, 5OP $P < 0.0001$, MOVA $P = 0.0014$; Ndufs1 BHIBA $P = 0.0069$, 5OP $P = 0.0023$, MOVA $P = 0.0003$). **q** $^{18}$F-FDG PET/CT of BHIBA-, 5OP-, and MOVA-treated mice identifies tissue-specific metabolic effects (representative images from five independent repeats). Scale bars are standardized uptake value (SUV) in rainbow scale (0 violet–5 red). Right—CT, middle—PET, left—PET/CT. $^{18}$F-FDG uptake into **r** BAT (BHIBA $P = 0.045$, MOVA $P = 0.016$), and **s** hind limb muscle (BHIBA $P = 0.043$, 5OP $P = 0.041$, MOVA $P = 0.0008$) and **t** forelimb muscle (5OP $P = 0.014$, MOVA $P < 0.0001$) of BHIBA-, 5OP-, and MOVA-treated mice ($n = 5$; two-tailed $t$-test). Dark green = MOVA, light green = 5OP, dark blue = BHIBA. Data in bar graphs are mean ± SEM with individual data points shown. Box and whisker plots show 25th to 75th percentile (box) min to max (whiskers), mean (+) and median (−). ∗$P \leq 0.05$, ^$P \leq 0.01$, ●$P \leq 0.001$, ‡$P \leq 0.0001$. Source data are provided as a Source Data file.

communicate redox state and, amongst other effects, recruit additional beige adipocytes to rescue systemic redox stress.

We acknowledge limitations to our study. Our work does not preclude the presence of other metabolite factors, protein signals, or bioactive lipids that may contribute to the functional signaling between adipose–adipose and adipose–skeletal muscle. Indeed, our data suggests the presence of an, as yet unidentified, secreted peptidic inhibitor of browning. Due to the importance of the biosynthetic enzymes involved in the generation of BHIBA, BHIVA, 5OP, and MOVA, shared across multiple metabolic pathways including BCAA catabolism, fatty acid oxidation, TCA cycle, glutamine/glutamate, and redox glutathione metabolism, directly targeting these in loss-of-function experiments would be incapable of unambiguously distinguishing the effect of the metabolites from perturbations of multiple pathways. However, this observation supports our finding that these metabolites function as key brown and beige adipokines, as the need to closely integrate signals influencing systemic energy balance with pathways responding to cellular energy and redox metabolism would be essential. Our analyses suggest that MOVA and 5OP function through cAMP–PKA–p38 MAPK signaling, and BHIBA functions through mTOR, to induce adipocyte and myocyte metabolic gene expression. Both cAMP–PKA–p38 MAPK and mTOR signaling regulate the activation of BAT thermogenesis, WAT browning, and mitochondrial biogenesis[30,53–55]. Our study suggests that BHIBA and MOVA function through extracellular receptors and that 5OP induces effects through a direct metabolic mechanism to induce adipocyte browning and myocyte β-oxidation. Future work may also uncover the effects of BHIBA, 5OP, and MOVA on other tissues, and the identity of the discreet receptors through which the metabokines exert their effects.

In addition, we ascertained that the metabolite BHIVA demonstrated some capacity to regulate adipose tissue metabolism in vitro, although no effect on energy or glucose homeostasis was observed in vivo. BHIVA has been shown to have bioactive properties in other settings, including skeletal muscle protein synthesis and exercise[56], and our work does not preclude an alternative signaling role for this metabolite linking beige adipose tissue and BAT to the regulation of systemic physiology.

We identify MOVA, 5OP, and BHIBA as a discrete set of small molecule brown and beige adipose metabokines. The identification of these metabolites as interorgan signals has significant implications, not only for our understanding of the integration and regulation of physiological energy metabolism and its protective role against the development of metabolic diseases, but also for understanding the pathophysiology of, and potential therapeutics for obesity, T2DM, and the metabolic syndrome.

## Methods

**Culture and differentiation of mouse primary adipocytes.** Primary white adipose stromal vascular cells were fractionated from 6- to 10-week-old C57BL/6J male mice as previously described[34,57]. Stromal vascular cells were then cultured and differentiated into adipocytes according to published methods[34,58]. During the 6-day differentiation, cells were cultured with 100 nM GW0742 (Sigma Aldrich), 250 nM HIC (Sigma Aldrich), 10 μM AKV (Sigma Aldrich), 5 μM AHI (Sigma Aldrich), 20 μM MOVA (Sigma Aldrich), 20 μM 5OP (Sigma Aldrich), 20 μM BHIBA (Sigma Aldrich), and 10 μM BHIVA (Sigma Aldrich) or with 1 μM forskolin (Sigma Aldrich) on days 5–6 of differentiation.

For media transfer experiments, treatment media was removed, cells were washed three times with phosphate-buffered saline (PBS), and 1.1-ml serum-free media was conditioned on fully differentiated primary white adipocytes that had remained untreated (control) or had received either 100 nM GW0742 throughout differentiation or 1 μM forskolin on days 5–6 of differentiation within each well of a 12-well plate. Media was removed after 24 hrs. Media was boiled at 100 °C for 5 mins for denaturation experiments. Conditioned media (1 ml) was added to naïve fully differentiated primary adipocytes in each well of a 12-well plate.

For assessment of aqueous fraction bioactivity and metabolomic analysis, after 24 h conditioning on cells, serum-free media was removed from the cells and spun at 600 g for 2 min to remove debris. Aqueous soluble metabolites were extracted from 1 ml of conditioned media as previously described[59]. For aqueous fraction bioactivity assessment, dried aqueous fractions were then reconstituted in 1 ml of fresh serum-free media by vortex-mixing (5 min) followed by sonication (15 min) and then added to naïve primary adipocytes in a 12-well plate for 24 h.

Mouse primary brown preadipocytes were isolated from C57BL6/J mice, cultured and differentiated according to established protocols[60].

**Human primary adipocyte culture.** Human white primary preadipocytes (PromoCell, Heidelberg, Germany, Cat no. C-12735) were seeded (10,000–15,000 cells/cm$^2$) and grown to confluence (37 °C, 5% CO$_2$) in preadipocyte growth medium (0.05 ml/ml fetal calf serum, 0.004 ml/ml endothelial cell growth supplement,

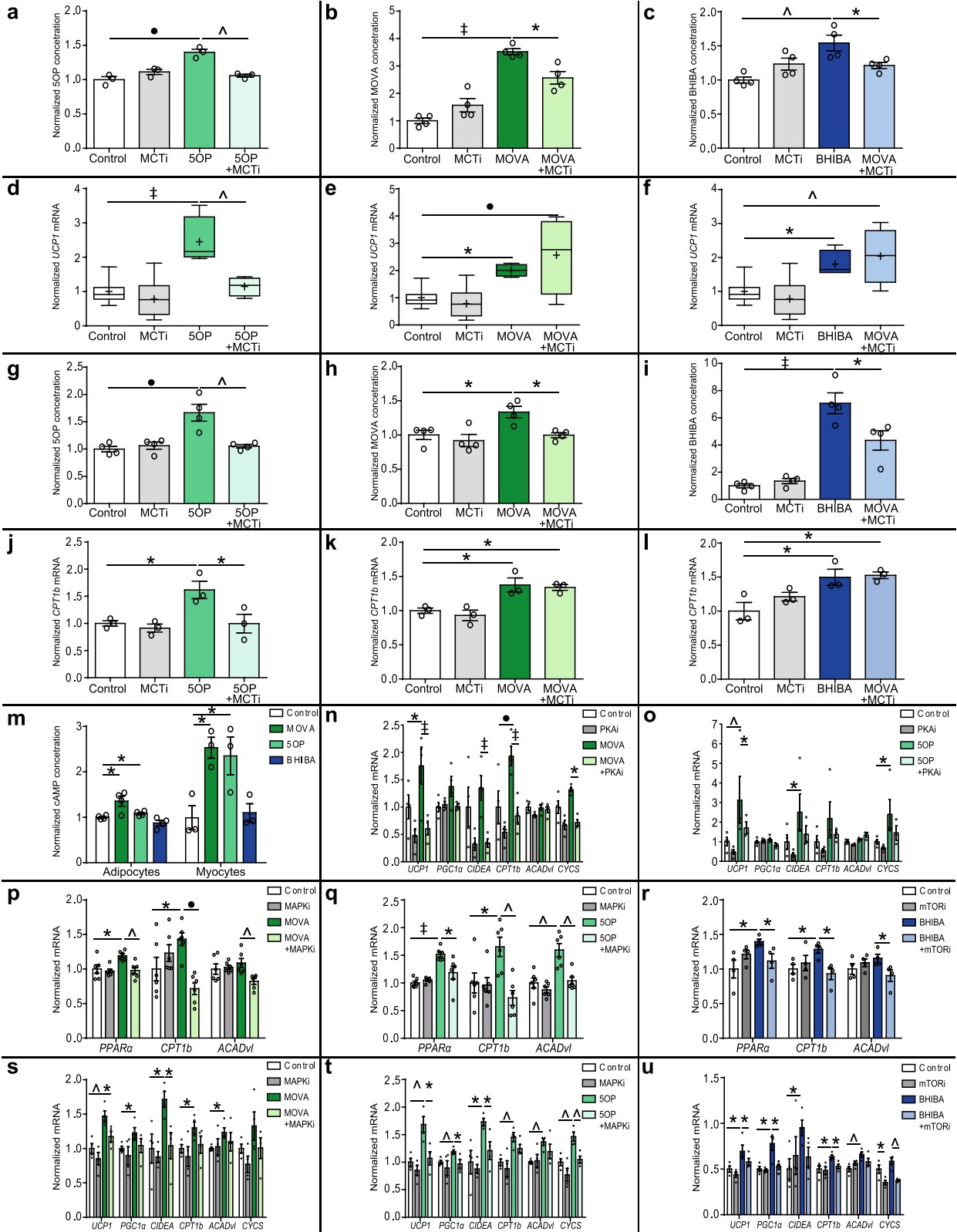

10 ng/ml epidermal growth factor, 1 μg/ml hydrocortisone, 90 μg/ml heparin). Preadipocytes were differentiated according to supplier's instructions. Briefly, growth medium was replaced by differentiation media (8 μg/ml d-biotin, 0.5 μg/ml insulin, 400 ng/ml dexamethasone, 44 μg/ml isobutylmethylxanthine, 9 ng/ml L-thyroxine, 3 μg/ml ciglitazone) for 48 h. Differentiation medium was replaced with adipocyte nutrition media (0.03-ml/ml fetal calf serum, 8-μg/ml d-biotin, 0.5-μg/ml insulin, 400-ng/ml dexamethasone) for 12 days until cells were fully differentiated. Media was changed every 48 h.

Human white primary adipocytes were treated throughout differentiation with either 100 nM GW0742, MOVA (10, 20, 40 μM), 5OP (10, 20, 40 μM), BHIBA (10, 20, 40 μM), and BHIVA (5, 10 μM), with 1 μM forskolin on days 10–12 of differentiation and with either 2 mM α-cyano-4-hydroxycinnamate for 24 h on day 11 of differentiation, or 500 nM of the p38 MAPK inhibitor Birb 796, 500 nM of the mTOR inhibitor temsirolimus, or 10 μM of the PKA inhibitor H89 for 12 h on day 12 of differentiation. All experiments were performed using adipocytes (day 12) at passages 3–5.

**Fig. 8 MOVA and 5OP signal through cAMP–PKA–p38 MAPK and BHIBA via mTOR.** The intracellular concentration of **a** 5-oxoproline (5OP) ($n = 3$, Control vs. 5OP $P = 0.0004$, 5OP vs. 5OP + MCTi $P = 0.0011$), **b** 3-methyl-2-oxovaleric acid (MOVA) ($n = 4$, Control vs. MOVA $P < 0.0001$, MOVA vs. MOVA + MCTi $P = 0.011$), and **c** β-hydroxyisobutyric acid (BHIBA) ($n = 4$, Control vs. BHIBA $P = 0.002$, BHIBA vs. BHIBA + MCTi $P = 0.05$) in human primary adipocytes treated concomitantly with the monocarboxylate transporter inhibitor α-cyano-4-hydroxycinnamate (MCTi) (2 mM), which prevents cellular export and import of monocarboxylate species (Two-way ANOVA with Tukey's post hoc). The expression of *UCP1* in human primary adipocytes treated concomitantly with MCTi and **d** 5OP (20 μM) (Control vs. 5OP $P < 0.0001$, 5OP vs. 5OP + MCTi $P = 0.0034$), **e** MOVA (20 μM) (Control vs. MOVA $P = 0.042$, Control vs. MOVA + MCTi $P = 0.0009$), and **f** BHIBA (20 μM) (Control vs. BHIBA $P = 0.047$, Control vs. BHIBA + MCTi $P = 0.007$) (Control $n = 12$; MCTi $n = 11$; 5OP, MOVA, BHIBA $n = 4$; 5OP + MCTi, MOVA + MCTi, BHIBA + MCTi $n = 4$; Two-way ANOVA with Tukey's post hoc). The intracellular concentration of **g** 5OP (Control vs. 5OP $P = 0.001$, 5OP vs. 5OP + MCTi $P = 0.0019$), **h** MOVA (Control vs. MOVA $P = 0.029$, MOVA vs. MOVA + MCTi $P = 0.027$), and **i** BHIBA (Control vs. BHIBA $P < 0.0001$, BHIBA vs. BHIBA + MCTi $P = 0.017$) in human primary skeletal myocytes treated concomitantly with the MCTi (2 mM) ($n = 4$; Two-way ANOVA with Tukey's post hoc). The expression of *CPT1b* in human primary myocytes treated concomitantly with MCTi and **j** 5OP (20 μM) (Control vs. 5OP $P = 0.035$, 5OP vs. 5OP + MCTi $P = 0.034$), **k** MOVA (20 μM) (Control vs. MOVA $P = 0.023$, Control vs. MOVA + MCTi $P = 0.038$), and **l** BHIBA (20 μM) (Control vs. BHIBA $P = 0.026$, Control vs. BHIBA + MCTi $P = 0.019$) ($n = 3$; Two-way ANOVA with Tukey's post hoc). **m** Normalized intracellular cAMP concentration measured by liquid chromatography–mass spectrometry in human primary adipocytes and myocytes treated with MOVA (20 μM), 5OP (20 μM), or BHIBA (20 μM) (Two-tailed *t*-test; adipocytes $n = 4$, MOVA $P = 0.02$, 5OP $P = 0.039$; myocytes $n = 3$, MOVA $P = 0.012$, 5OP $P = 0.05$). Human primary adipocytes treated with the selective protein kinase A inhibitor H89 (10 μM) (PKAi) with and without **n** 20 μM MOVA (*UCP1* MOVA vs. Control $P = 0.004$, MOVA vs. MOVA + PKAi $P < 0.0001$; *CIDEA* MOVA vs. MOVA + PKAi $P = 0.0001$; *CPT1b* MOVA vs. Control $P = 0.0003$, MOVA vs. MOVA + PKAi $P < 0.0001$; *CYCS* MOVA vs. MOVA + PKAi $P = 0.029$) and **o** 20 μM 5OP (*UCP1* 5OP vs. Control $P = 0.0019$, 5OP vs. 5OP + PKAi $P = 0.036$; *CIDEA* 5OP vs. Control $P = 0.03$; *CYCS* 5OP vs. Control $P = 0.05$) ($n = 4$; Two-way ANOVA with Holm-Sidak post hoc). Human primary myocytes treated with the p38 MAPK inhibitor Birb 796 (500 nM) (p38 MAPKi) with and without **p** 20 μM MOVA ($n = 6$; Two-tailed *t*-test; Control vs. MOVA *PPARα* $P = 0.017$, *CPT1b* $P = 0.05$; MOVA vs. MOVA + MAPKi *PPARα* $P = 0.003$, *CPT1b* $P = 0.0002$, *ACADvl* $P = 0.007$) and **q** 20 μM 5OP ($n = 6$; Two-tailed *t*-test; Control vs. 5OP *PPARα* $P < 0.0001$, *CPT1b* $P = 0.026$, *ACADvl* $P = 0.0024$; 5OP vs. 5OP + MAPKi *PPARα* $P = 0.022$, *CPT1b* $P = 0.002$, *ACADvl* $P = 0.0021$). **r** Human primary myocytes treated with the mTOR inhibitor temsirolimus (500 nM) (mTORi) with and without 20 μM BHIBA ($n = 4$; Two-tailed *t*-test; Control vs. BHIBA *PPARα* $P = 0.026$, *CPT1b* $P = 0.017$; BHIBA vs. BHIBA + mTORi *PPARα* $P = 0.044$, *CPT1b* $P = 0.012$, *ACADvl* $P = 0.05$). Human primary adipocytes treated with the p38 MAPK inhibitor Birb 796 (500 nM) (p38 MAPKi) with and without **s** 20 μM MOVA ($n = 4$; Two-tailed *t*-test; Control vs. MOVA *UCP1* $P = 0.0041$, *PGC1α* $P = 0.037$, *CIDEA* $P = 0.025$, *CPT1b* $P = 0.029$, *ACADvl* $P = 0.017$; MOVA vs. MOVA + MAPKi *UCP1* $P = 0.033$, *CIDEA* $P = 0.023$) and **t** 20 μM 5OP ($n = 4$; Two-tailed *t*-test; Control vs. 5OP *UCP1* $P = 0.0052$, *PGC1α* $P = 0.007$, *CIDEA* $P = 0.016$, *CPT1b* $P = 0.0045$, *ACADvl* $P = 0.0013$, *CYCS* $P = 0.0047$; 5OP vs. 5OP + MAPKi *UCP1* $P = 0.017$, *PGC1α* $P = 0.041$, *CIDEA* $P = 0.024$, *CYCS* $P = 0.0068$). **u** Human primary adipocytes treated with mTORi (500 nM) with and without 20 μM BHIBA ($n = 4$; Two-tailed *t*-test; Control vs. BHIBA *UCP1* $P = 0.044$, *PGC1α* $P = 0.013$, *CIDEA* $P = 0.018$, *CPT1b* $P = 0.025$, *ACADvl* $P = 0.007$; BHIBA vs. BHIBA + mTORi *UCP1* $P = 0.047$, *PGC1α* $P = 0.019$, *CPT1b* $P = 0.014$, *CYCS* $P = 0.0014$; Control vs. mTORi *CYCS* $P = 0.029$). Dark green = MOVA, light green = 5OP, dark blue = BHIBA. Data in bar graphs are mean ± SEM with individual data points shown. Box and whisker plots show 25th to 75th percentile (box) min to max (whiskers), mean (+) and median (−). *$P \le 0.05$, ^$P \le 0.01$, ●$P \le 0.001$, ‡$P \le 0.0001$. Source data are provided as a Source Data file.

Informed consent was obtained from donors. The cells were approved and complied with ethics according to:

– Collection, generation, research purpose, and sale in compliance with the Declaration of Helsinki: PromoCell GmbH Sickingenstr. 63/65 69126 Heidelberg Germany.
– Use in compliance with Human Tissue Act (UK) by Leeds Institute of Cardiovascular and Metabolic Medicine, University of Leeds, Leeds, LS2 9JT UK in 2017.

**siRNA MCT1 knockdown**. FlexiTube siRNA against MCT1 (SI04330396), AllStars negative control siRNA, and HiPerFect Transfection Reagent were purchased from Qiagen. Adipocyte transfection was performed according to the manufacturer's instructions (75 ng siRNA, 3 μL transfection reagent per well, 60 nmol/L final siRNA concentration) on days 0, 5, and 10 of differentiation.

**C2C12 myocytes**. C2C12 myocytes were cultured as previously described[61]. C2C12 cells (Sigma Aldrich Cat no. 91031101) were cultured in 1 mL Dulbecco's Modified Eagle's Medium (DMEM; 4.5 g/L glucose, L-glutamine, NaHCO3, and pyridoxine-HCl) supplemented with 10% fetal bovine serum and 1× penicillin/streptomycin (100 U/mL and 100 μg/mL, respectively) in 12-well plates (Millipore, USA) at 37 °C in 5% $CO_2$. Once confluent, cells were maintained in differentiation medium for 6 days (DMEM supplemented with 2% horse serum and 1× penicillin/streptomycin). Medium was changed daily.

**Human primary skeletal myocyte culture**. Adult human skeletal myoblasts (Cell applications Inc. Cat no. 150-05a) were grown in human skeletal muscle cell growth medium (Cell applications Inc.) at 37 °C within a humidified atmosphere at 5% $CO_2$. Subculture of human skeletal myoblasts occurred once 85–95% confluency was reached. Experiments were limited to the 5th passage. Human skeletal myoblasts were seeded at 9500 cells per cm$^2$. Once confluent, myoblasts were cultured for 6–8 days in skeletal muscle differentiation media (Cell applications Inc.) to induce myoblast differentiation to myotubes. Myoblasts were treated with human skeletal muscle cell differentiation medium (Cell applications Inc.) containing either BHIBA (5 and 20

μM), MOVA (5 and 20 μM), 5OP (5 and 20 μM), BHIVA (2.5 and 10 μM), α-cyano-4-hydroxycinnamate (24 h on day 8, 2 mM), p38 MAPK inhibitor Birb 796 (12 h on day 8, 500 nM) (Sigma), or temsirolimus (12 h on day 8, 500 nM).

Informed consent was obtained from donors. The cells were approved and complied with ethics according to:

– Collection, generation, research purpose, and sale: Cell Applications, Inc. 5820 Oberlin Dr. Suite 101, San Diego, CA 92121.
– Use in compliance with Human Tissue Act (UK) by Leeds Institute of Cardiovascular and Metabolic Medicine, University of Leeds, Leeds, LS2 9JT UK in 2017.

**Immortalized human white preadipocyte culture**. Immortalized human white preadipocytes, isolated from human neck fat, were obtained from Yu-Hua Tseng (Joslin Diabetes Center, Harvard Medical School, USA)[62]. Cells were cultured according to a previously published protocol[62]. Cells were treated throughout induction and maintenance phases with either 20 μM MOVA, 20 μM 5OP, 20 μM BHIBA, or 10 μM BHIVA or for the final 4 days of differentiation with 1 μM forskolin.

Informed consent was obtained from donors. The cells were approved and complied with ethics according to:

– Generation and use including drug discovery purposes: Joslin Diabetes Center, One Joslin Place, Boston, MA 02215, USA in 2015.
– Use in compliance with Human Tissue Act (UK) by MedImmune (AstraZeneca) Aaron Klug Building, Granta Park, Cambridge, CB21 6GH, UK in 2016.

**Gene expression array**. Total RNA was extracted and reverse transcribed into cDNA using ambion TaqMan Gene Expression Cells-to-$C_T$ kits (AM1729). cDNA was preamplified using TaqMan PreAmp MasterMix Kits (Applied Biosystems 4384267), prior to loading on TaqMan OpenArray Real-Time PCR Plates (Applied Biosystems 4406947) and analysis by RT-PCR on a QuantStudio 12K Flex RT-PCR System using QuantStudio software version 1.2.10 (ThermoFisher).

Data analysis was carried out in the ThermoFisher Cloud Relative Quantification App (3.4.1-PCR-build3 2017-09-26). Data were normalized to PPIA, RPLP0, and B2M. Primer details are given in Supplementary Table 9.

**Confocal images and image quantification.** Cells were washed with PBS and labeled with 1:500 HCS Lipidtox Green (ThermoFisher, H34475) and 1:10,000 Hoechst 33342 (ThermoFisher, H3570) at 37 °C for 30 min. Afterwards, cells were washed three times with PBS, fixed in 3.7% formaldehyde for 15 min, and washed four times with PBS. Cells were resuspended in donkey blocking buffer (1xPBS, 5% donkey serum, 0.3% Triton X-100) and incubated at 4 °C overnight. Blocking buffer was removed prior to addition of 10-μg/ml mouse anti-human/anti-mouse UCP1 monoclonal antibody (R&D Systems MAB6158, Minneapolis, USA) in antibody buffer (1xPBS, 1% BSA, 0.3% Triton X-100) for 1 h at 21 °C. Cells were washed three times with PBS, 3-μg/ml Alexa455 AffiniPure donkey anti-mouse IgG (H + L) antibody (Jackson ImmunoResearch 715-545-150, West Grove, USA) in antibody buffer (in 1xPBS, 1% BSA, 0.3% Triton X-100) was added for 1 h at 21 °C and washed three times with PBS. Samples were afterwards kept in PBS.

Imaging was performed on an OPERA QEHS spinning disc confocal microscope (PerkinElmer) using an ×20-magnification water immersion lens (numerical aperture 0.7), and 405, 488, and 561-nm lasers with appropriate filter sets. Image analysis was performed using Columbus software version 2.9.1 (PerkinElmer). To determine UCP1 protein levels, first, areas with differentiated adipocytes were defined using LipidTox labeling of lipid droplets followed by measurement of mean UCP1 intensity levels within this region.

**Fatty acid and glucose uptake assays.** The cellular fatty acid uptake assay was performed as described in the literature[35].

The cellular glucose uptake assay was performed as follows. Cells were grown and differentiated in 96-well plates. Cells were washed with Dulbecco's phosphate-buffered saline (DPBS) and placed in low glucose (1 g/L) serum-free DMEM for 24 h. Media was aspirated and replaced with fresh low glucose serum-free DMEM for 1 h. Following media aspiration DPBS containing 6-deoxy-6-[(7-nitro-2,1,3-benzoxadiazol-4-yl)amino]-D-glucose (200 μM) was added for 1 h and cells were kept at 37 °C, 5% $CO_2$. Cells were washed three times with DPBS and fluorescence measured using a microplate reader (excitation 485 nm, emission 528 nm).

**[13]C-Palmitate substrate labeling study.** Palmitate was solubilized using a dialyzed albumin solution[59]. [13]C-Palmitate labeling studies were performed as previously described[58]. Briefly, fully differentiated human primary white adipocytes were conditioned with serum-free medium containing insulin 850 nmol/L, triiodothyronine 1 nmol/L, and rosiglitazone 1- and 140-μmol/L U-[13]C–labeled palmitate (Cambridge Isotope Laboratories). After 24 h, cells were collected and metabolites extracted as previously described. During differentiation, cells were cultured with either 20 μM MOVA, 20 μM 5OP, 20 μM BHIBA, or 10 μM BHIVA.

**[13]C-labeled amino acid substrate analysis.** Fully differentiated human primary white adipocytes were conditioned with serum-free medium containing insulin 850 nmol/L, triiodothyronine 1 nmol/L, and rosiglitazone 1 μmol/L and either 200 μmol/L U-[13]C–labeled leucine, 200 μmol/L U-[13]C-labeled isoleucine, 200 μmol/L U-[13]C-labeled valine or 100 μmol/L U-[13]C-labeled glutamate (Cambridge Isotope Laboratories). After 24 h, cells were collected and metabolites extracted as previously described. During differentiation, cells were cultured with 1 μM forskolin on days 10–12 of differentiation. Extracted metabolites were analyzed by GC-MS as described below and previously[58,59].

**Animal experimentation.** Six-week-old C57BL6/J mice (Charles River) were weight-matched and assigned to groups for treatment. Mice were treated with either 100 mg/kg/day 5OP, 100 mg/kg/day MOVA, 150 mg/kg/day BHIBA, or 125 mg/kg/day BHIVA in their drinking water for 17 weeks and fed either standard chow or a 60% fat-diet ad libitum (Bio Serv F3282). Animals were housed in conventional cages at room temperature with humidity maintained at 40–60% and a 12-h light/dark photoperiod.

The cold exposure study was conducted at the University of Cambridge. Animals were housed in a specific pathogen-free facility with 12-h light and dark cycles and humidity maintained at 40–60%. Four groups of eight C57BL6/J mice (Charles River) underwent thermal adaptation at 12 weeks of age. One group was placed at 8 °C for 4 weeks, a second group was maintained at room temperature for 3 weeks then placed at 8 °C for 1 week, a further group was placed at 28 °C for 4 weeks and a final group was maintained at room temperature for 4 weeks (21–23 °C). All mice were killed at 16 weeks of age. This study was regulated under the Animals (Scientific Procedures) Act 1986 Amendment Regulations 2012 following ethical review by the University of Cambridge Animal Welfare and Ethical Review Body.

All studies complied with national and local ethical regulations for animal research. All procedures were carried out in accordance with U.K. Home Office protocols under a U.K. Home Office Project License by a U.K. Home Office Personal License Holder.

**Indirect calorimetry.** All experiments were performed according to previously published protocols[34]. Briefly Combined Laboratory Animal Monitoring System (CLAMS) (Columbus Instruments) was used to monitor oxygen consumption, carbon dioxide production, food intake, and locomotory activity using Oxymax software (version 5.37.05, Columbus Instruments). The CLAMS was calibrated before each experiment. Animals were subjected to a 3-day acclimation period in a training cage to habituate to the environment of the metabolic cages. Animals were maintained in normal bedding at 22 °C throughout the monitoring period. Ten minute interval measurements for each animal were obtained for oxygen and carbon dioxide with ad libitum access to food and water (or water plus metabolites) on a controlled 12-h light/dark cycle. Cages contained one mass sensor to monitor food intake. Data were analyzed using CaIR (version 1.1) (https://calrapp.org/)[63].

**Intraperitoneal glucose tolerance tests (IPGTTs).** IPGTTs were performed as previously described[34]. Mice were fasted for 8 h with free access to water prior to baseline glucose measurements. Administration of glucose (Sigma Aldrich) was performed by intraperitoneal injection (glucose 1.5 mg/g of body weight; glucose solution 150 mg/ml). Blood was obtained from the tail vein immediately prior to glucose injection and then at 30, 60, 90, and 120 min post injection. Glucose levels were measured using a Bayer Contour Glucose Meter (Bayer Healthcare).

**Positron emission tomography/computed tomography (PET/CT).** PET/CT scans were performed on an Albira Si (Bruker). Mice were anaesthetized under 2–3% isoflurane, weighed and injected intravenously with 8.9 ± 2.3-MBq [18]F-FDG in 200 μL via the lateral tail vein and flushed with saline, followed by a 1-h uptake period. Mice were scanned for 20-min static PET, followed by a 10-min CT protocol for anatomical registration. The CT scans were performed at a 35-kV tube voltage and 200 μA over 250 projections. Animal temperature was maintained and monitored throughout the procedure alongside the respiratory rate. PET/CT data were reconstructed using the Albira Reconstructor Software in PMOD (version 3.807, Bruker). PET data were reconstructed using a maximum likelihood expectation maximization iterative method at 25 iterations with scatter, random event, and radiotracer decay corrections. The PET data were fused with the CT data, which was reconstructed with filtered back projection. All PET and CT image data were analyzed in PMOD (version 3.807, Bruker). The methods used to calculate adiposity have been described previously[64].

**Blood and tissue collection.** Mice were killed by cervical dislocation. Blood was obtained by cardiac puncture, collected in tubes containing EDTA (2.5 mmol/L), and immediately centrifuged to obtain plasma. WAT, interscapular BAT, soleus, and gastrocnemius muscle were removed and flash-frozen in liquid nitrogen.

**Gene expression analysis.** Total RNA extraction from WAT, adipocytes, and myocytes; cDNA conversion; and quantitative RT-PCR were performed according to published protocols[34]. All data were normalized to 18S rRNA (human WAT, adipocytes, and myocytes; human *18SrRNA* primer PPH05666E-200, mouse *18SrRNA* primer PPM57735E-200, Qiagen) and quantitative measures were obtained using the ΔΔCT method. Data were analyzed using StepOne™ Software (version 2.1 Applied Biosystems). Details of primers are given in Supplementary Table 10.

**RNA-Seq.** Next-generation RNA sequencing was performed by Cambridge Genomic Services (Cambridge UK). The Lexogen 3′mRNA seq kit (illumina) was used for library preparation. Samples were analyzed using a NextSeq500 (illumina) with 75 bp per read and 10 million reads per sample. Data analysis was performed using R package edgeR v3.8.6.

**Protein analysis.** Analysis of UCP1, PGC-1α, CPT1, and NDUFS1 was performed using ELISA per the manufacturer's instructions (UCP1 Kit SEF557Ra, PGC-1α Kit SEH337Ra; CPT1 SEF360Mu; NDUFS1 SEJ794Hu Cloud-Clone Corp., Houston, TX). Kinase profiling was performed using the Proteome Profiler Phospho-MAPK Array Kit (Bio-techne Ltd; ARY002B) according to the manufacturer's instructions.

**Citrate synthase assay.** Citrate synthase was assayed according to published protocols[58]. Cell and tissue samples were homogenized in 100 mM $K_2HPO_4$/$KH_2PO_4$, 5 mM EDTA, 0.1-mM fructose-2,6-bisphosphate, 0.1% Triton X-100, and 1 mM dithiothreitol, pH 7.2. Citrate synthase activity was measured at 412 nm to detect the transfer of sulfhydryl groups to 5,5′-dithiobis(2-nitrobenzoic acid) (DTNB). Reaction buffer composition was 100 mM Tris · HCl, 0.2 mM acetyl CoA, 0.1 mM DTNB, and 1 mM oxaloacetate (omitted for control), pH 8.0. The reaction rates were linear for ≥4 min. All assays were run in duplicate, and means were analyzed. Specific activities were expressed in international units (μmol substrate transformed to product/min) normalized to tissue weight.

**Histology and immunohistochemistry.** Tissue was fixed in 4% paraformaldehyde, processed in paraffin, and sectioned into 4-μm sections for staining. Sections were

deparaffinized in xylene, rehydrated through a 95–50% ethanol series, then placed in water before staining. The VECTASTAIN Elite ABC HRP Kit (Vector Labs, cat no: PK 6100) was used as per manufacturer's instructions to retrieve antigens. Anti-UCP1 primary antibody (abcam ab23841) was used at a 1:100 dilution and incubated on sections in a humidity chamber at room temperature for 60 min. Visualization of VECTASTAIN peroxidases was achieved using VECTOR NovaRED Peroxidase (HRP) Substrate Kit (Vector Labs, cat no: SK-4800). Sections were rinsed in tap water and counterstained with hematoxylin (VECTOR Hematoxylin QS, Vector Labs, Cat no: H-3404). Sections were rinsed in tap water, dehydrated through a 50–95% ethanol series, cleared in xylene, and mounted onto coverslips using DPX. Pictures were captured using a standard light microscope and Zen 2 pro software (Version 2.0, ZEISS).

**Respirometry**. Basal respiration was measured in adipocytes and skeletal myocytes maintained in Krebs-Henseleit buffer at 37 °C using Clark-type oxygen electrodes (Strathkelvin Instruments, Strathkelvin, U.K.) and software Si 782 System (version 4.1 Strathkelvin Instruments) as described previously[58]. In addition, respiration was also measured in human primary adipocytes following the addition of succinate (20 mM).

Mitochondrial respiration profiles of immortalized human white preadipocytes isolated from neck fat and differentiated to mature adipocytes in the presence of MOVA (20 μM), 5OP (20 μM), BHIBA (20 μM), or BHIVA (10 μM) were assessed on a Seahorse XFe96 Flux Analyzer using Seahorse Wave Software (version 2.6.1) (Agilent Technologies, Waldbronn, Germany), using Agilent MitoStress kits according to manufacturer's instructions. Briefly, immortalized human white preadipocytes were seeded into XFe96 cell culture plates and cultured as described earlier. Oxygen consumption rates (OCR) were monitored during the sequential injection of oligomycin (1 μM, ATP-Synthase coupled respiration), carbonyl-cyanide-4-(trifluoromethoxy)-phenylhydrazone (FCCP) (1 μM, maximal respiration), and rotenone/antimycin A (0.5 μM each, non-mitochondrial respiration). OCR were afterwards normalized to cell number as determined by Hoechst 33342 staining and measurement on a Cytation 5 (BioTek, Winooski, USA) with DAPI filter.

High-resolution respirometry was carried out on human primary skeletal myoblasts differentiated to mature myotubes in the presence of MOVA (20 μM) or 5OP (20 μM) using an Oxygraph-2k (Oroboros, Innsbruck). Cells were suspended in 2.4 mL MiR05 (MiR05: 110 mM sucrose, 60 mM K-lactobionate, 20 mM HEPES, 20 mM taurine, 10 mM KH2PO4, 3 mM MgCl2, 0.5 mM EGTA, 1% (w/v) fatty acid-free BSA, pH 7.1). One hundred microliter of cell suspension was used to determine cell counts using a Millipore Scepter Handheld Automated Cell Counter. Respiration rates of myocytes were measured using Clark-type polarographic oxygen electrodes (Oxygraph-2k; Oroboros, Innsbruck) maintained at 37 °C under a constant stir speed of 500 rpm. Cells were permeabilized with digitonin (Caymann Chemicals, 2.5 μg/mL) immediately prior to adding the suspension into the Oxygraph chamber. Substrates and inhibitors were added to the chamber and steady rates of respiration recorded. Substrates were added to the chamber in the following order giving the final concentrations in the chamber indicated; Malate 2 mM; Octanoyl-Carnitine (APExBIO Technology) 0.4 mM; ADP 10 mM; pyruvate 5 mM; glutamate 10 mM; succinate 10 mM; CCCP 0.5 μM; rotenone 0.25 μM. Respiration rates were corrected for cell number. Data were processed using DatLab (version 6.1, Oroboros Instruments).

**Tissue and primary cell metabolite extraction**. Metabolites were extracted from WAT (30 mg), BAT (20 mg), skeletal muscle (20 mg), and primary skeletal myocytes and adipocytes and cell culture media (500 μl) as previously described[59]. Methanol–chloroform (2:1, 600 μl) was added to the samples. Tissue and cells were homogenized in a TissueLyser II (Qiagen) (2 min 25 Hz) and the samples were sonicated for 15 min. Chloroform–water (1:1) was then added (200 μl of each). Samples were centrifuged (16,100 g, 20 min) and the aqueous phases were separated and stored at −80 °C until analysis.

**GC-MS metabolomic analysis**. Dried aqueous phase samples were derivatized using methoxyamine hydrochloride solution (20 mg/ml in pyridine; Sigma Aldrich) and 30 μl of N-methyl-N-trimethylsilyltrifluoroacetamide (Macherey-Nagel, Duran, Germany) as described previously[59]. GC-MS metabolomics and data analysis were performed according to published methods[59]. All GC-MS analyses were made using a Trace GC Ultra coupled to a Trace DSQ II single-quadrupole mass spectrometer (Thermo Scientific, Cheshire, UK). Derivatized aqueous samples were injected splitless onto a 30 m × 0.25 mm 5% phenylpolysilphenylene-siloxane column with a 0.25-μm ZB-5 ms stationary phase (Phenomenex). The injector temperature was 230 °C, and the helium carrier gas was used at a flow rate of 1.2 ml/min. The initial column temperature of 70 °C was increased by 10 °C/min to 130 °C and then increased at a rate of 5 °C/min to 230 °C followed by an increase of 20 °C/min to 310 °C and held for 5 min (transfer line temperature = 250 °C). Trace DSQ II single-quadrupole mass spectrometer (Thermo Scientific, Cheshire, UK) was operated with an ion source temperature of 250 °C and electron ionization energy of 70 eV with positive ionisation. The detector was turned on after 240 s, and full-scan spectra were collected using 3 scans/s over a range of 50–650 m/z.

GC-MS chromatograms were processed using Xcaliber (version 2.2; Thermo Scientific). Each individual peak was integrated and then normalized. Overlapping peaks were separated using traces of single ions. Peak assignment was based on mass fragmentation patterns matched to the National Institute of Standards and Technology (USA) library, and to previously reported literature and was confirmed using standards.

**LC-MS metabolomic analysis**. LC-MS metabolomic analysis was performed using a QTRAP 4000 quadrupole mass spectrometer (AB Sciex, Warrington, UK) coupled to an Acquity ultra performance liquid chromatography system from Waters Ltd. (Atlas Park, Manchester, UK) according to a previously described method[34,58,65]. cAMP was measured as previously described[58]. Chromatography was performed on an Atlantis HILIC silica 3 μm 2.1 × 150-mm column (Waters Corporation) at 30 °C using 0.1% formic acid and 10 mM ammonium formate as solvent A and 0.1% formic acid in acetonitrile as solvent B, at 250 uL/min across a 32-min chromatography run time. The 4000 QTRAP tandem mass spectrometer was operated with a Turbo V electrospray source. The mass spectrometer was operated in positive multiple reaction monitoring (MRM) mode with the following mass spectrometric parameters: curtain gas 20 psi, source temperature 450 °C, ion source gas 1 30 psi, ion source gas 2 40 psi, interface heater ON, collision gas high, ionspray voltage 5000 eV, entrance potential 10 eV, resolution quadrupole 1 unit, resolution quadrupole 2 unit, MRM window 70 msec, target scan time 1 s. All metabolite concentrations were determined using the standard addition method[34]. LC-MS chromatograms were processed, and individual peaks integrated and normalized using Analyst (version 1.6.2; AB Sciex).

**Human adipose biopsies**. Forty-two eligible and consecutive patients undergoing routine de novo pacemaker implantation at Leeds General Infirmary volunteered to participate in the study and provided written consent (Supplementary Table 7). Human subcutaneous adipose tissue biopsies were obtained prior to any instrumentation of the central circulation. Lidocaine was initially injected to anaesthetize the area and a small incision was made under the left clavicle. Immediately prior to creation of the pre-pectoral pocket for the generator above the pectoralis major a portion of subcutaneous adipose tissue (~100 mg) was sampled. This sample was immediately snap-frozen in liquid nitrogen. Following this, pacemaker procedures were otherwise completed routinely. There were no complications ascribed to the sampling procedure. The study is approved by the Leeds West Research Ethics Committee (11/YH/0291) and Leeds Teaching Hospitals Trust R&D committee (CD11/10015) and conforms to the Declaration of Helsinki.

**Human GWAS database analysis**. The Type 2 Diabetes Knowledge Portal (http://www.type2diabetesgenetics.org/) was searched for genetic variants in the genes encoding the metabokine biosynthetic enzymes (BCAT2, BCKDHB, ACADS, HADHA, GSS, GGCT, and SLC16A1) and BMI.

**Statistical analysis**. Multivariate data analysis was performed using SIMCA-P+ 13.0 (Umetrics AB, Umeå, Sweden) and Metaboanalyst 4.0 (https://www.metaboanalyst.ca/MetaboAnalyst/Secure/analysis/AnalysisView.xhtml)[66]. GC-MS and LC-MS data sets were scaled to unit variance. Data sets were analyzed using PLS-DA. Metabolite changes responsible for clustering or regression trends within the pattern recognition models were identified by interrogating the corresponding loadings plot. Metabolites identified in the variable importance in projections/coefficients plots were deemed to have changed globally if they contributed to separation in the models with a confidence limit of 95%. $Q^2$ values are shown as a measure of model robustness.

Indirect calorimetry data were analyzed, and $P$ values were calculated using ANCOVA/Generalized Linear Model with body mass as a covariate in CaIR (version 1.1, https://calrapp.org/)[63].

All other univariate data, unless otherwise stated, are expressed as means, and error bars depict SEM. A two-tailed Student's $t$ test, Benjamini Hochberg-adjusted $t$-test, Exact test, One or Two-way ANOVA were used to determine $P$ value set as a nominal significance of $P \leq 0.05$ with multiple comparisons correction using a Dunnett's, Holm-Sidak, or Tukey's post hoc test as indicated. Univariate statistics was conducted using Prism (version 6.02, Graphpad).

**Reporting summary**. Further information on research design is available in the Nature Research Reporting Summary linked to this article.

## Data availability

RNA-Seq data associated with this study are available from Gene Expression Omnibus [GSE129153] (https://www.ncbi.nlm.nih.gov/geo/query/acc.cgi?acc=GSE129153). Mass spectrometry metabolomics data associated with this study have been deposited to EBI MetaboLights Database [MTBLS2436]. In addition, all identified metabolites with identifying information are presented in the Source data provided with this paper. Human GWAS data are available from The Type 2 Diabetes Knowledge Portal (http://www.type2diabetesgenetics.org/).

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

## Acknowledgements

L.D.R. acknowledges the support of the Diabetes UK RD Lawrence Fellowship (16/0005382), the Biotechnology and Biological Research Council (BB/R013500/1), and the Medical Research Council (MR/R014086/1). J.L.G. is supported by grants from the Medical Research Council (MC_UP_A090_1006; MC_PC_13030; MR/P011705/1 and MR/P01836X/1) and the Biotechnology and Biological Research Council (BB/H013539/2). We thank Yu-Hua Tseng from Harvard University for provision of immortalized human primary adipocytes. RNA-seq analysis was performed by Cambridge Genomic Services.

## Author contributions

A.W. and A.M. were directly involved in the majority of experiments. F.N.K. and C.C. performed and analyzed immortalized human adipocyte studies. F.N.K. performed mass spectrometry cAMP assays in mouse tissues. J.L.S, A.M., and E.B. assisted with experiments throughout. S.A.M., B.D.M., L.D.R. and J.L.G. assisted with metabolomic screens and $^{13}$C isotope studies. G.R.D. and J.D. assisted with quantitative microscopy and open array analysis. A.J.M. and A.D.V.M. assisted with the design and performance of respirometry. A.D.V.M. assisted with myocyte culture studies. S.V. and A.V.-P. led and designed the mouse cold acclimatization studies. J.W. and J.E.S. ran the PET/CT imaging studies. J.G. and K.K.W. isolated the human adipose biopsies and plasma samples. L.D.R. designed and led the studies, interpreted the results, and wrote the paper with input from all co-authors.

## Competing interests

The authors declare no competing interests.
