## [Peer Review File · Nature Communications]

Reviewers' Comments:

Reviewer #2:

Remarks to the Author:

The quantity of data in the paper has increased, aiming to answer the reviewers' requests, including a new claim, arguing for MCTs as transporters of the investigated metabolites. Reading this diverse manuscript, some major concerns still remain, concerning transport, respirometry and indirect calorimetry. Please see for details below.

Transport:

Overall, the new data provide good hints that MCTs are involved, as intracellular levels of the metabolites increase in the presence of an inhibitor, while extracellular levels decrease. However, one would usually prefer a clean transport assay to demonstrate the transport of the metabolites. If I would have to accept the MCT transport claim, could the authors include a canonical MCT substrate in their metabolite analysis, such as lactate? Furthermore, the claim that this is specifically the MCT1, that is solely based on correlative nature and should be flagged as such. It would require at least one more independent method, such as genetic manipulation of MCT1 levels or proteoliposomal transport assays, to really manifest this claim.

Figure 7 provides indirect evidence for 5-oxoproline import but are the other tested metabolites also internalized? Given a dominant role of lactate in thermogenic adipocytes and browning (as seen also in Carrière et al. 2014, Diabetes), is the lactate transport competitive with the newly identified metabolites? Furthermore, why were muscle cells not treated with the MCT inhibitor, serving as an independent cell type that is important for this paper?

Respirometry:

There is some discrepancy between methodology and results. The methods describe that the Seahorse was used for immortalized adipocytes, using the manufacturer's protocol. In the new Figure 2 I –o, I only see the basal and leak respiration. What happened to the maximal respiration rates using FCCP? Please provide the missing parameter, and in the supplements, the original Seahorse traces rather than only the bar charts in the main figure.

Also concerning respirometry, the 'substrate-inhibitor high-resolution respirometry protocol' appears to some extent a bit useless (Fig. 4 e-g). How do the authors de-convolute glutamate from octanoyl-carnitine oxidation rates (as fatty oxidation could easily impose slower electron flux rates in the ETC), and how, in the absence of rotenone, is oxaloacetate accumulation prevented (which would inhibit complex 2 activity)? As a notion, what is referred to as maximal ETC capacity, that is actually maximal substrate oxidation rate (with the given substrates). What is 'uncoupling ratio' in Fig. 4g (which is by the way not referred to in the main text)? Is this respiratory control ratio as established by Mitchell and colleagues, and nicely described by Nicholls/Ferguson or Brand/Nicholls, or a new expression from an instrument manual? How is this parameter calculated?

Indirect Calorimetry:

It is great that the authors use ANCOVA with body mass as covariate and have consulted the established relevant literature for interpretation of mouse energy metabolism. There was actually no need to summarize it. However, if this literature had been fully embraced, the authors would not suggest to illustrate weight corrected data for clarity. The argument for ANCOVA, in length re-

cited in the responses, would be completely useless by adding this illustration.

As a minor add-on, I thought this study aimed to delineate obese vs non-obese, and not larger vs smaller, as stated in the rebuttal. If I have misunderstood this, please provide body length data of the mice to substantiate that one group of mice is smaller than the other.

Finally, I accept the overall concept of the study on metabolite signaling. In this study, however, there is bias on the adipose tissue-muscle axis. The reader is still left in the dark, possibly asking whether other tissues (e.g. heart, liver, muscle) would, in response to the identical stimuli, release exactly the same metabolites? And why would these metabolites not act on other tissues, which also possess the MCT? Can this bias be excluded?

Reviewer #3:

Remarks to the Author:

The authors have addressed my comments.

Reviewer #4:

Remarks to the Author:

The revised version has added a large amount of data supporting the notion that the three brown/beige adipocytes-secreted metabolites (MOVA, 5OP and BHIBA) regulate systemic energy metabolism through inter-organ crosstalk. The authors provided additional evidence showing that MOVA and 5OP signal through cAMP-PKA-p38 MAPK and BHIBA via mTOR.

How do MOVA and 5OP induce intracellular cAMP level in brown adipocytes and myocytes? Do these two metabolites also increase cAMP in hepatocytes. If so, what are the metabolic effects of MOVA and 5OP in hepatocytes?

Response to Reviewers - Brown and beige adipose tissue regulate systemic metabolism to resist diet-induced obesity through metabolite signals in an inter-organ signaling axis – 25628-A

Reviewer #2:

The quantity of data in the paper has increased, aiming to answer the reviewers' requests, including a new claim, arguing for MCTs as transporters of the investigated metabolites.

Reading this diverse manuscript, some major concerns still remain, concerning transport, respirometry and indirect calorimetry. Please see for details below.

Transport:

Overall, the new data provide good hints that MCTs are involved, as intracellular levels of the metabolites increase in the presence of an inhibitor, while extracellular levels decrease. However, one would usually prefer a clean transport assay to demonstrate the transport of the metabolites. If I would have to accept the MCT transport claim, could the authors include a canonical MCT substrate in their metabolite analysis, such as lactate? Furthermore, the claim that this is specifically the MCT1, that is solely based on correlative nature and should be flagged as such. It would require at least one more independent method, such as genetic manipulation of MCT1 levels or proteoliposomal transport assays, to really manifest this claim.

We thank the reviewer for their comment. Although lactate is widely considered a canonical substrate for MCTs (MCTs 1-4), and is quantitatively the most prominent, it is well established that MCTs 1-4 physiologically transport a wide variety of molecules including pyruvate, ketone bodies beta-hydroxybutyrate (to which BHIBA is closely structurally related) and acetoacetate, and the ketoacid metabolite class (to which BHIVA and MOVA belong). These substrate specificities have been reviewed extensively¹⁻⁴. Moreover 5OP, MOVA and BHIVA are transported by MCT1^{3,5-7}. Therefore our intention was not specifically to show that MCTs transport our metabolite signals through kinetic or transporter assays, for which there is already evidence in the literature, but that MCTs are involved in the forskolin-induced browning-mediated secretion of the metabokines from adipocytes. We have clarified this in the manuscript (**page 7, paragraph 3**):

"5OP, MOVA and BHIVA are transported by MCT1^{3,5-7} MCTs also transport the BHIBA structurally-related ketone body beta-hydroxybutyrate^{4,5}."

"We used a pharmacological MCT inhibitor (MCTi; α -cyano-4-hydroxycinnamate) to determine the involvement of MCTs in browning-mediated secretion of the metabokines".

We greatly appreciate the reviewer's comment regarding MCT specificity. Although we did not directly claim that the transport was solely mediated by MCT1 in our previous version of the manuscript, we have performed additional experiments to strengthen the evidence for the involvement of MCT1 in the forskolin-induced browning-mediated secretion of the metabokines from adipocytes. We followed the reviewer's suggestion and employed siRNA to knockdown the expression of MCT1 by 88% in human primary adipocytes. Knockdown of MCT1 phenocopied the effects of the inhibitor studies, inhibiting forskolin-induced secretion of the metabokine signals, decreasing MOVA, 5OP, BHIBA and BHIVA extracellular concentration whilst increasing their intracellular concentration (**page 7, paragraph 3**):

"To confirm MCT1 contributed to the browning-mediated secretion of the metabokines from adipocytes, we decreased MCT1 expression in human adipocytes by 88% using siRNA (Fig 3a). Knockdown of MCT1 inhibited forskolin-induced export of the metabolites from adipocytes, again decreasing the metabolite extracellular concentration whilst increasing their intracellular concentration (Fig 3b – e)."

Figure 7 provides indirect evidence for 5-oxoproline import but are the other tested metabolites also internalized? Given a dominant role of lactate in thermogenic adipocytes and browning (as seen also in Carrière et al. 2014, Diabetes), is the lactate transport competitive with the newly identified metabolites? Furthermore, why were muscle cells not treated with the MCT inhibitor, serving as an independent cell type that is important for this paper?

We now include new data on the intracellular concentration of 5OP, MOVA and BHIBA in human primary adipocytes treated with the metabokines with and without co-treatment with the MCT inhibitor (**page 14, paragraph 2, Fig 8 a-c**):

"Treatment of human adipocytes with MOVA, 5OP or BHIBA increased the intracellular concentration of the metabokines; this effect was abrogated by co-treatment with the MCTi (α -cyano-4-hydroxycinnamate) (Fig 8a-c)"

In addition, as the reviewer suggests, we have performed new experiments to determine whether the metabokines function extracellularly or intracellularly at the human skeletal myocyte. We identify that, as in adipocytes, inhibiting MCTs in myocytes impairs the increase in metabokine concentrations observed following treatment with the metabolites (**page 15, paragraph 1, Fig 8g-i**):

“We then examined whether the metabokines signaled via similar mechanisms in human primary skeletal myocytes. Treatment of human skeletal myocytes with the metabokines increased their intracellular concentration; this effect was impaired by co-treatment with the MCTi (Fig 8g-i).”

We have also further defined the mechanisms through which the metabolites signal in myocytes. Using dual treatment of myocytes with the metabokines and the monocarboxylate transporter inhibitor, we identify that 5-oxoproline regulates expression of CPT1b in human myocytes through a mechanism that requires internalisation into the cell. In contrast, we find that BHIBA and MOVA do not require import into the myocyte to elicit their effects on CPT1b expression and likely signal through receptors on the myocyte surface (**page 15, paragraph 2, Fig 8j-l**):

“Combined MCTi and 5OP treatment impaired 5OP-mediated CPT1b expression in myocytes (Fig 8j). The MOVA and BHIBA-induced expression of CPT1b was not impaired by MCTi (Fig 8k & l). These data suggest that 5OP requires import into the cells to induce molecular signals leading to increased metabolic gene expression. Conversely these results indicate MOVA and BHIBA function through extracellular signal transduction and may require a receptor in the adipocyte and myocyte membrane.”

Respirometry:

There is some discrepancy between methodology and results. The methods describe that the Seahorse was used for immortalized adipocytes, using the manufacturer’s protocol. In the new Figure 2 l –o, I only see the basal and leak respiration. What happened to the maximal respiration rates using FCCP? Please provide the missing parameter, and in the supplements, the original Seahorse traces rather than only the bar charts in the main figure.

We originally focussed on basal and leak respiration rates as these are the most relevant parameters of the manufacturer’s protocol for physiological uncoupling as occurs through the action of UCP1. We now include chemically-uncoupled maximal respiration as requested (**Fig 2l-o**) and mitostress assay traces (**Supplementary Fig 2n – q**) (**page 6, paragraph 2**):

“Functionally, leak respiration and electron transport chain uncoupling were increased in immortalized human adipocytes treated with the metabolites (Fig 2l-o) (Supplementary Fig. 2n - q). Adipocyte chemically-uncoupled maximal respiration was also increased by MOVA and 5OP (Fig 2l&m).”

Also concerning respirometry, the ‘substrate-inhibitor high-resolution respirometry protocol’ appears to some extent a bit useless (Fig. 4 e-g). How do the authors deconvolute glutamate from octanoyl-carnitine oxidation rates (as fatty oxidation could easily impose slower electron flux rates in the ETC), and how, in the absence of rotenone, is oxaloacetate accumulation prevented (which would inhibit complex 2 activity)?

We thank the reviewer for highlighting the weakness with our previous substrate-inhibitor high-resolution respirometry protocol. As 5OP and MOVA were found to induce expression of fatty acid β -oxidation genes in both murine and human myocytes we have performed an alternative experiment using a protocol designed firstly to evaluate fatty acid β -oxidation supported respiration in permeabilized primary human myocytes treated with MOVA and 5OP. This protocol uses substrates/inhibitors in the following order: Malate/Octanoyl-carnitine; ADP; Pyruvate/Glutamate; Succinate; CCCP; Rotenone. Our analyses identify MOVA and 5OP increase fatty acid oxidation supported respiration in human primary skeletal myocytes and that 5OP increases chemically-uncoupled maximal substrate oxidation (**page 8, paragraph 3, Fig. 4e & f**):

“MOVA and 5OP induced the greatest increase in fatty acid oxidation gene expression in both mouse and human primary myocytes. These metabolites were selected for characterization in primary myocytes using a substrate-inhibitor high-resolution respirometry protocol. MOVA and 5OP increased respiratory capacity in permeabilized human myocytes supported by substrates for fatty acid β -oxidation (octanoyl-carnitine/malate/ADP) (Fig 4e & f). 5OP also increased chemically-uncoupled maximal substrate oxidation (CCCP) in myocytes (Fig 4f).”

As a notion, what is referred to as maximal ETC capacity, that is actually maximal substrate oxidation rate (with the given substrates).

Given the reviewer’s suggestion we now refer to this parameter as “chemically-uncoupled maximal substrate oxidation” (**page 9; paragraph 1**):

5OP also increased chemically-uncoupled maximal substrate oxidation (CCCP) in myocytes (Fig 4f).”

What is ‘uncoupling ratio’ in Fig. 4g (which is by the way not referred to in the main text)? Is this respiratory control ratio as established by Mitchell and colleagues, and nicely described by Nicholls/Ferguson or Brand/Nicholls, or a new expression from an instrument manual? How is this parameter calculated?

We have removed the uncoupling ratio data as it was relevant to the previously presented data which has also been removed from the manuscript and replaced with alternative data as described above.

Indirect Calorimetry:

It is great that the authors use ANCOVA with body mass as covariate and have consulted the established relevant literature for interpretation of mouse energy metabolism. There was actually no need to summarize it. However, if this literature had been fully embraced, the authors would not suggest to illustrate weight corrected data for clarity. The argument for ANCOVA, in length re-cited in the responses, would be completely useless by adding this illustration.

As a minor add-on, I thought this study aimed to delineate obese vs non-obese, and not larger vs smaller, as stated in the rebuttal. If I have misunderstood this, please provide body length data of the mice to substantiate that one group of mice is smaller than the other.

We agree with the reviewer, however we have found that the use of ANCOVA in the analysis of indirect calorimetry data is not embraced by all. We have removed the weight corrected data as the reviewer suggests. In addition, we apologize for our lack of clarity in our previous response, as the reviewer understands our aim was to delineate metabolic parameters between treated mice (lower adiposity and body weight) and control mice (higher adiposity and body weight), as described in the manuscript, and not body length.

Finally, I accept the overall concept of the study on metabolite signaling. In this study, however, there is bias on the adipose tissue-muscle axis. The reader is still left in the dark, possibly asking whether other tissues (e.g. heart, liver, muscle) would, in response to the identical stimuli, release exactly the same metabolites? And why would these metabolites not act on other tissues, which also possess the MCT? Can this bias be excluded?

We thank the reviewer for highlighting this point. However, we would respectfully counter that the focus on adipose and skeletal muscle in this paper is not intrinsic bias but aligns to our carefully considered hypothesis and is founded in the current literature, which supports the presence of adipose-adipose and adipose-skeletal muscle signalling axes. We outline some of this supporting literature in our introduction (**page 3, paragraph 2**):

“Transplantation studies of both beige and brown fat in mice suggest that these tissues can signal to activate endogenous beige and brown fat and improve glucose homeostasis in

skeletal muscle^{8,9}. *In murine models of both adipose tissue browning and increased BAT thermogenesis, fatty acid oxidation in skeletal muscle is increased*⁸⁻¹⁰.”

For these reasons, as with the majority of inter-organ signal discoveries in the literature, in this manuscript we focussed on interaction between a discreet set of tissues (adipose and skeletal muscle). It is possible that the metabokines outlined in this manuscript may form signaling axes between other tissues in defined physiological settings (again as many inter-organ cytokines, metabokines and lipokines have been found to do in subsequent publications following their discovery). Exploring every potential tissue in a single study or manuscript is unfeasible. We have certainly considered that the metabolites may have functional effects on other tissues, as we state within our discussion (**page 19, paragraph 1**):

“Future work may also uncover the effects of BHIBA, 5OP and MOVA on other tissues, and the identity of the discreet receptors through which the metabokines exert their effects.”

In addition, the reviewer mentions “And why would these metabolites not act on other tissues, which also possess the MCT?” We would like to highlight that only 5OP was found to require MCT-mediated transport into adipocytes and myocytes as a target tissue to induce expression of UCP1 and CPT1b, respectively.

Although outside the remit of this current manuscript, we appreciate that investigations into the signaling mechanisms of BHIBA, 5OP and MOVA in, and effects on, other tissues will be an important topic for future studies. While tangential to the current study hypothesis, in this response, we provide data below to further illustrate our points above. We examine the concentration of our candidate molecules in the liver of cold conditioned study mice:

We examined the liver of mice housed at thermoneutrality, room temperature and under thermogenic conditions with cold exposure at 8°C for a period of one week and one month. The metabolites MOVA and BHIVA were not detected in the livers of these mice. The hepatic concentrations of 5OP and BHIBA were not significantly different between mice housed at thermoneutrality, room temperature and with one week or one month cold exposure:

Cold exposure does not increase liver concentrations of the metabokines

GC-MS analysis of MOVA, 5OP, BHIVA and BHIBA concentration in the liver of mice housed at thermoneutrality (TN), room temperature (RT), 8°C for 1 week (W) or 8°C for 1 month (M).

These data are highly consistent with the finding that branched-chain amino acids (precursors to BHIVA, BHIBA and MOVA) are poorly metabolized in the liver as the liver expresses low levels of the mitochondrial branched chain aminotransferases (BCAT1 and BCAT2), the first enzymes in the catabolism of BCAAs^{11,12}. Whereas adipose tissue branched chain amino acid metabolism directly modulates circulating branched chain-amino acid concentrations¹³. This highlights the important point that, as with other inter-organ signals, a functional level of expression of the signals (or their biosynthetic enzymes in this case) is required in the tissue of origin and imparts a great deal of specificity. We highlight the importance of the regulation of the metabokine biosynthetic pathways in adipocytes in the main manuscript using ¹³C tracer experiments in adipocytes treated with forskolin (**page 6-7, Supplementary Fig 3**). Therefore this data, presented for the reviewer, suggests that other tissues, in this case liver, do not generate the same molecules with the same physiological stimuli.

We have also performed additional analyses to investigate the effect of the metabokines on hepatic metabolism. As reported in the manuscript, MOVA, 5OP and BHIBA increased systemic energy expenditure in six-week-old mice fed standard chow and either MOVA (100 mg/kg/day), 5OP (100 mg/kg/day) or BHIBA (150 mg/kg/day) in drinking water for 17 weeks compared to control mice. MOVA, 5OP and BHIBA increased expression of fatty acid β -oxidation and mitochondrial genes in brown and white adipose tissue and skeletal muscle. Within the manuscript we describe that MOVA and 5OP mediate these effects through cyclic AMP (cAMP). Therefore using the liquid chromatography method described within the

manuscript we analysed the cAMP concentration in the livers of our metabokine treated mice. Hepatic cAMP concentration was not altered by treatment with the metabokines:

The metabokines MOVA, 5OP and BHIBA do not increase hepatic cyclic AMP *in vivo*

Normalized cyclic AMP concentrations in the liver of mice treated with 100 mg/Kg/day MOVA, 100 mg/Kg/day 5OP or 125 mg/Kg/day BHIBA for 17 weeks (n = 5).

The expression of a panel of mitochondrial and fatty acid β -oxidation genes in the livers of MOVA, 5OP and BHIBA-treated mice was not significantly different from control:

RT-qPCR Expression of mitochondrial and metabolic genes in the liver of metabokine-treated mice. Data are Mean \pm SEM (n = 5).

In addition, citrate synthase activity (as a proxy for mitochondrial number) was not significantly different from control mice in the livers of MOVA, 5OP and BHIBA treated mice:

Mitochondrial content determined by citrate synthase assay is not increased in the livers of metabokine treated mice. Data are mean \pm SEM (n = 4).

Therefore, the metabokines are not increased by cold exposure in the liver and the metabolites do not regulate hepatic cAMP concentrations, the expression of hepatic β -oxidation genes or mitochondrial biogenesis in the liver. Given the already data heavy manuscript and that this additional data is outside the remit, focus and hypothesis of this current study, focussed on the cross-talk between fat tissues and skeletal muscle, we present this data here for peer review purposes.

Reviewer #3 (Remarks to the Author):

The authors have addressed my comments

We thank the reviewer for their input which has greatly strengthened the manuscript.

Reviewer #4 (Remarks to the Author):

The revised version has added a large amount of data supporting the notion that the three brown/beige adipocytes-secreted metabokines (MOVA, 5OP and BHIBA) regulate systemic energy metabolism through inter-organ crosstalk. The authors provided additional evidence showing that MOVA and 5OP signal through cAMP-PKA-p38 MAPK and BHIBA via mTOR.

How do MOVA and SOP induce intracellular cAMP level in brown adipocytes and myocytes?

We thank the reviewer for their comments regarding the nature of the signalling mechanisms in myocytes. We have performed new experiments to determine whether the metabokines signal intracellularly or extracellularly at the human primary myocyte to induce their effects. Using dual treatment of human primary skeletal myocytes with both the metabokines and MCT inhibitor we identify that, as in adipocytes, inhibiting MCTs in myocytes impairs the increase in metabokine concentrations observed following treatment with the metabolites (**page 15, paragraph 1, Fig 8g-i**):

“We then examined whether the metabokines signaled via similar mechanisms in human primary skeletal myocytes. Treatment of human skeletal myocytes with the metabokines increased their intracellular concentration; this effect was impaired by co-treatment with the MCTi (Fig 8g-i).”

Using dual treatment of myocytes with the metabokines and the monocarboxylate transporter inhibitor, we identify that 5-oxoproline regulates expression of CPT1b in human myocytes through a mechanism that requires internalisation into the cell. In contrast, we find that BHIBA and MOVA do not require import into the myocyte to elicit their effects on CPT1b expression and likely signal through receptors on the myocyte surface (**page 15, paragraph 1, Fig 8j-l**):

“Combined MCTi and 5OP treatment impaired 5OP-mediated CPT1b expression in myocytes (Fig 8j). The MOVA and BHIBA-induced expression of CPT1b was not impaired by MCTi (Fig 8k&l). These data suggest that 5OP requires import into the cells to induce molecular signals leading to increased metabolic gene expression. Conversely these results indicate MOVA and BHIBA function through extracellular signal transduction and may require a receptor in the adipocyte and myocyte membrane.”

In addition we now include data showing that 5OP and MOVA increase cAMP concentrations in human primary myocytes (**page 15, paragraph 3, Fig 8m**):

“Using LC-MS we measured the intracellular cAMP content in human adipocytes and myocytes treated with 5OP, MOVA and BHIBA (Fig 8m)”

Do these two metabokines also increase cAMP in hepatocytes. If so, what are the metabolic effects of MOVA and 5OP in hepatocytes?

We thank the reviewer for highlighting potential effects of the metabokines on hepatocytes. We have considered that the metabolites may have functional effects on other tissues besides adipose tissue and skeletal muscle, as we state within our discussion (**page 19, paragraph 1**):

“Future work may also uncover the effects of BHIBA, 5OP and MOVA on other tissues, and the identity of the discreet receptors through which the metabokines exert their effects.”

However, we would like to emphasize that the focus on adipose and skeletal muscle in this paper aligns to the hypothesis which we set out to test and is founded in the current literature which supports the presence of adipose-adipose and adipose-skeletal muscle signalling axes. We outline some of this supporting literature in our introduction (**page 3, paragraph 2**):

“Transplantation studies of both beige and brown fat in mice suggest that these tissues can signal to activate endogenous beige and brown fat and improve glucose homeostasis in skeletal muscle^{8,9}. In murine models of both adipose tissue browning and increased BAT thermogenesis, fatty acid oxidation in skeletal muscle is increased⁸⁻¹⁰.”

Although outside the remit of this current manuscript, we appreciate that investigations into the effects of BHIBA, 5OP and MOVA on other tissues will be an important topic for future studies. Therefore, in this response, we provide data below examining the effect of the metabokines on hepatic metabolism. As reported in the manuscript, MOVA, 5OP and BHIBA increased systemic energy expenditure in six-week-old mice fed standard chow and either MOVA (100 mg/kg/day), 5OP (100 mg/kg/day) or BHIBA (150 mg/kg/day) in drinking water for 17 weeks compared to control mice. MOVA, 5OP and BHIBA increased expression of fatty acid β -oxidation and mitochondrial genes in brown and white adipose tissue and skeletal muscle. Within the manuscript we describe that MOVA and 5OP mediate these effects through cAMP. Therefore using the liquid chromatography method described within the manuscript we analysed the cAMP concentration in the livers of our metabokine treated mice. Hepatic cAMP concentration was not altered by treatment with the metabokines:

The metabokines MOVA, 5OP and BHIBA do not increase hepatic cyclic AMP *in vivo*

Normalized cyclic AMP concentrations in the liver of mice treated with 100 mg/Kg/day MOVA, 100 mg/Kg/day 5OP or 125 mg/Kg/day BHIBA for 17 weeks (n = 5).

The expression of a panel of mitochondrial and fatty acid β -oxidation genes in the livers of MOVA, 5OP and BHIBA-treated mice was not significantly different from control:

RT-qPCR Expression of mitochondrial and metabolic genes in the liver of metabokine-treated mice. Data are Mean \pm SEM (n = 5).

In addition, citrate synthase activity (as a proxy for mitochondrial number) was not significantly different from control mice in the livers of MOVA, 5OP and BHIBA treated mice:

Mitochondrial content determined by citrate synthase assay is not increased in the livers of metabokine treated mice. Data are mean +/- SEM (n = 4).

Therefore, the metabolites do not regulate hepatic cAMP concentrations, the expression of hepatic β -oxidation genes or mitochondrial biogenesis in the liver. Given the already data heavy manuscript and that this additional data is outside the scope, focus and hypothesis of this current study, focussed on the cross-talk between fat tissues and skeletal muscle, we present this data here for peer review purposes.

References

1. Halestrap, A.P. & Price, N.T. The proton-linked monocarboxylate transporter (MCT) family: structure, function and regulation. *Biochem J* **343 Pt 2**, 281-299 (1999).
2. Halestrap, A.P. & Meredith, D. The SLC16 gene family-from monocarboxylate transporters (MCTs) to aromatic amino acid transporters and beyond. *Pflugers Arch* **447**, 619-628 (2004).
3. Halestrap, A.P. The monocarboxylate transporter family--Structure and functional characterization. *IUBMB Life* **64**, 1-9 (2012).
4. Halestrap, A.P. The SLC16 gene family - structure, role and regulation in health and disease. *Mol Aspects Med* **34**, 337-349 (2013).
5. Sasaki, S., Futagi, Y., Kobayashi, M., Ogura, J. & Iseki, K. Functional characterization of 5-oxoproline transport via SLC16A1/MCT1. *J Biol Chem* **290**, 2303-2311 (2015).
6. Silva, L.S., *et al.* Branched-chain ketoacids secreted by glioblastoma cells via MCT1 modulate macrophage phenotype. *EMBO Rep* **18**, 2172-2185 (2017).
7. Ogura, J., *et al.* Transport Mechanisms for the Nutritional Supplement beta-Hydroxy-beta-Methylbutyrate (HMB) in Mammalian Cells. *Pharm Res* **36**, 84 (2019).
8. Tran, T.T., Yamamoto, Y., Gesta, S. & Kahn, C.R. Beneficial effects of subcutaneous fat transplantation on metabolism. *Cell Metab* **7**, 410-420 (2008).
9. Liu, X., *et al.* Brown adipose tissue transplantation improves whole-body energy metabolism. *Cell research* **23**, 851-854 (2013).
10. Kong, X., *et al.* Brown Adipose Tissue Controls Skeletal Muscle Function via the Secretion of Myostatin. *Cell Metab* (2018).
11. Ichihara, A. & Koyama, E. Transaminase of branched chain amino acids. I. Branched chain amino acids-alpha-ketoglutarate transaminase. *J Biochem* **59**, 160-169 (1966).

12. Wahren, J., Felig, P. & Hagenfeldt, L. Effect of protein ingestion on splanchnic and leg metabolism in normal man and in patients with diabetes mellitus. *J Clin Invest* **57**, 987-999 (1976).
13. Herman, M.A., She, P., Peroni, O.D., Lynch, C.J. & Kahn, B.B. Adipose tissue branched chain amino acid (BCAA) metabolism modulates circulating BCAA levels. *J Biol Chem* **285**, 11348-11356 (2010).

Reviewers' Comments:

Reviewer #2:

Remarks to the Author:

The authors have invested major efforts to take this study as far as they could, making it suitable for publication, at least from my point of view. I have no further major comments.

Minor:

1. Lines 153-157 - this is up to the authors: the difference between leak respiration and electron chain uncoupling may not be understandable for all readers, especially as 'electron chain uncoupling' cannot be found in figure and caption. A more logical way to describe these findings would be to state that 'basal and leak respiration are both increased, partially due to increased electron flux seen as chemically-uncoupled maximal respiration, and partially due to increased proton conductance seen as decreased coupling efficiency.

2. I did not go through all the references in the introduction but found this paragraph with odd or not timely citations, up to the authors to adjust their citations:

55 The anti-obesity and

56 anti-diabetic effects of brown and beige adipose tissues are also not solely reliant on the
57 thermogenic process. Mice lacking UCP1 are resistant to diet-induced obesity at room
58 temperature, yet mice lacking brown/beige fat are highly susceptible to an obese
59 phenotype¹¹⁻¹⁵. Therefore, beige and brown fat may influence systemic metabolism through
60 non-UCP1 thermogenic mechanisms, potentially mediated through the release of endocrine
61 signals in the adipocyte "secretome".

I see that citation 12 fits nicely to the text, also cit 13, 14 and 15 fit to some extent but I do not understand how cit 11 fits in here, which describes the dependence on UCP1 at thermoneutrality. Furthermore, in lines 59-61, the authors imply that this UCP1-independent phenomenon is completely unresolved, omitting some findings in Nature Communications earlier this year (PMID: 32005798).

Reviewer #4:

Remarks to the Author:

The revised version has fully addressed my major concerns. Thank you.

Reviewer #2 (Remarks to the Author):

The authors have invested major efforts to take this study as far as they could, making it suitable for publication, at least from my point of view. I have no further major comments.

We thank the reviewer for their comments and recognition of our effort.

Minor:

1. Lines 153-157 - this is up to the authors: the difference between leak respiration and electron chain uncoupling may not be understandable for all readers, especially as ‘electron chain uncoupling’ cannot be found in figure and caption. A more logical way to describe these findings would be to state that ‘basal and leak respiration are both increased, partially due to increased electron flux seen as chemically-uncoupled maximal respiration, and partially due to increased proton conductance seen as decreased coupling efficiency.

We thank the reviewer for their suggestion. We have incorporated their suggested text into the manuscript (**page 6**):

“Functionally, basal and leak respiration are both increased in immortalized human adipocytes treated with the metabolites, partially due to increased electron flux seen as chemically-uncoupled maximal respiration, and partially due to increased proton conductance seen as decreased coupling efficiency (Fig 2l-o) (Supplementary Fig. 2n - q).”

2. I did not go through all the references in the introduction but found this paragraph with odd or not timely citations, up to the authors to adjust their citations:

The anti-obesity and anti-diabetic effects of brown and beige adipose tissues are also not solely reliant on the thermogenic process. Mice lacking UCP1 are resistant to diet-induced obesity at room temperature, yet mice lacking brown/beige fat are highly susceptible to an obese phenotype¹¹⁻¹⁵. Therefore, beige and brown fat may influence systemic metabolism through non-UCP1 thermogenic mechanisms, potentially mediated through the release of endocrine signals in the adipocyte “secretome”.

I see that citation 12 fits nicely to the text, also cit 13, 14 and 15 fit to some extent but I

do not understand how cit 11 fits in here, which describes the dependence on UCP1 at thermoneutrality.

We appreciate the reviewer's suggestion. This citation (11; "UCP1 Ablation Induces Obesity and Abolishes Diet-induced Thermogenesis in Mice Exempt from Thermal Stress by Living at Thermoneutrality") is less important for the point made in the text than the others noted by the reviewer (12-15). We can see how inclusion could lead to confusion. Therefore, we have opted to remove the citation from the paragraph on **page 3**.

Furthermore, in lines 59-61, the authors imply that this UCP1-independent phenomenon is completely unresolved, omitting some findings in Nature Communications earlier this year (PMID: 32005798).

As suggested by the reviewer, we have added the recommended citation to the text of the introduction (**page 3**):

"Therefore, beige and brown fat may influence systemic metabolism through non-UCP1 thermogenic mechanisms¹⁶, potentially mediated through the release of endocrine signals in the adipocyte secretome."

Reviewer #4 (Remarks to the Author):

The revised version has fully addressed my major concerns. Thank you.

We thank the reviewer for their comments which have substantially improved the manuscript.